# The active zone protein Clarinet regulates synaptic sorting of ATG-9 and presynaptic autophagy

Zhao Xuan[1☯¤], Sisi Yang[1☯], Benjamin Clark[1], Sarah E. Hill[1], Laura Manning[1], Daniel A. Colón-Ramos[1,2,3]*

**1** Program in Cellular Neuroscience, Neurodegeneration and Repair, Departments of Neuroscience and Cell Biology, Yale University School of Medicine, New Haven, Connecticut, United States of America, **2** Instituto de Neurobiología, Recinto de Ciencias Médicas, Universidad de Puerto Rico, San Juan, Puerto Rico, **3** Wu Tsai Institute, Yale University, New Haven, Connecticut, United States of America

☯ These authors contributed equally to this work.
¤ Current address: School of Biology and Ecology, University of Maine, Orono, Maine, United States of America
* daniel.colon-ramos@yale.edu

**Data Availability Statement:** All relevant data are within the paper and its Supporting Information files.

## Abstract

Autophagy is essential for cellular homeostasis and function. In neurons, autophagosome biogenesis is temporally and spatially regulated to occur near presynaptic sites, in part via the trafficking of autophagy transmembrane protein ATG-9. The molecules that regulate autophagy by sorting ATG-9 at synapses remain largely unknown. Here, we conduct forward genetic screens at single synapses of *C. elegans* neurons and identify a role for the long isoform of the active zone protein Clarinet (CLA-1L) in regulating sorting of autophagy protein ATG-9 at synapses, and presynaptic autophagy. We determine that disrupting CLA-1L results in abnormal accumulation of ATG-9 containing vesicles enriched with clathrin. The ATG-9 phenotype in *cla-1(L)* mutants is not observed for other synaptic vesicle proteins, suggesting distinct mechanisms that regulate sorting of ATG-9-containing vesicles and synaptic vesicles. Through genetic analyses, we uncover the adaptor protein complexes that genetically interact with CLA-1 in ATG-9 sorting. We also determine that CLA-1L extends from the active zone to the periactive zone and genetically interacts with periactive zone proteins in ATG-9 sorting. Our findings reveal novel roles for active zone proteins in the sorting of ATG-9 and in presynaptic autophagy.

## Introduction

Macroautophagy (herein called autophagy) is a well-conserved cellular degradative pathway, and its disruption in neurons results in axonal degeneration, accumulation of protein aggregates, and cell death [1–4]. Autophagy is spatially and temporally regulated in neurons, and autophagosome biogenesis occurs near presynaptic sites and in response to increased neuronal activity states [5–16]. Directed transport of autophagy proteins to presynaptic sites contributes to local autophagosome biogenesis at synapses [5,9,16–18]. How autophagy proteins are

**Funding:** SY was supported by China Scholarship Council-Yale World Scholars Program (no grant number)(https://medicine.yale.edu/bbs/training/initiatives/csc/). ZX, BC, SY, SEH, LM and DACR were supported by NIH R01NS076558, DP1NS111778 (https://www.nih.gov/) and by an HHMI Scholar Award 55108513 (https://www.hhmi.org/). The funders had no role in study design, data collection and analysis, decision to publish, or preparation of the manuscript.

**Competing interests:** The authors have declared that no competing interests exist.

**Abbreviations:** ADBE, activity-dependent bulk endocytosis; CME, clathrin-mediated endocytosis; EM, electron microscopy; L4, larva 4; SD, standard deviation; SNP, single-nucleotide polymorphism; TGN, *trans*-Golgi network; WGS, whole-genome sequencing.

transported, sorted, and locally regulated in neurons to control synaptic autophagy is not well understood.

ATG-9, a transmembrane protein necessary for autophagy, is actively trafficked in vesicles to promote local autophagosome biogenesis [19–24]. In neurons, ATG-9 is transported to presynaptic sites. Like synaptic vesicles, ATG-9-containing vesicles undergo activity-dependent exo-endocytosis regulated by canonical synaptic molecules such as endophilin A and synaptojanin 1 [24]. Activity-dependent exo-endocytosis of ATG-9-containing vesicles supports synaptic autophagy [24]. Yet, beyond the requirement of endophilin A and synaptojanin 1, the mechanisms that sort ATG-9 at presynaptic sites, the relationship of these mechanisms to those that sort canonical synaptic vesicle proteins and the relationship between ATG-9 and the synaptic machinery is not well understood. Knowledge of the molecules required for specific sorting of ATG-9 at synapses is of critical importance to understand the mechanisms that regulate synaptic autophagy.

To better understand the in vivo mechanisms that regulate ATG-9 sorting at synapses and their relationship with synaptic vesicle proteins, we generated transgenic *Caenorhabditis elegans* (*C. elegans*) strains that allowed us to simultaneously observe, in single neurons, localization of ATG-9 and synaptic vesicle proteins [24]. We performed unbiased forward genetic screens for mutants in which ATG-9 localization was differentially affected as compared to synaptic vesicle proteins, and identified an allele that affects the long isoform of the active zone protein Clarinet (CLA-1L). Clarinet bears similarity to *Drosophila* active zone protein Fife and vertebrate active zone proteins RIM, Piccolo, and Bassoon [25–29]. In *cla-1(L)* mutants ATG-9, but not synaptic vesicle proteins, abnormally accumulate to subsynaptic regions enriched for clathrin. This abnormal ATG-9 phenotype is suppressed by mutants for synaptic vesicle exocytosis, suggesting that the ATG-9 phenotype in *cla-1(L)* mutants emerges from defects in ATG-9 sorting during exo-endocytosis. Through genetic analyses, we found that mutants of the clathrin-associated adaptor complexes AP-2 and AP180 phenocopy and enhance the ATG-9 phenotypes observed for *cla-1(L)* mutant, whereas mutants for the AP-1 adaptor complex and the F-BAR protein syndapin 1 suppress the phenotype. We also observed that CLA-1L extends from the exocytic active zone to the endocytic periactive zone and genetically interacts with the periactive zone proteins EHS-1/EPS15 and ITSN-1/intersectin 1 in mediating ATG-9 sorting at presynaptic sites.

Our findings support a model whereby CLA-1L bridges the exocytic active zone regions with the endocytic periactive zone to regulate presynaptic sorting of ATG-9, likely via endosome-mediated sorting. Our findings also suggest that in vivo, ATG-9 containing vesicles represent a distinct subpopulation of synaptic vesicles. Our study uncovers molecules and synaptic machinery specifically involved in ATG-9 sorting and underscores the importance of active zone proteins in regulating local sorting of autophagy proteins and presynaptic autophagy.

## Results

### The active zone protein Clarinet (CLA-1) regulates ATG-9 sorting at presynaptic sites

To understand the in vivo mechanisms that regulate ATG-9 sorting at synapses and their relationship to the sorting of synaptic vesicle proteins, we simultaneously examined synaptic vesicle proteins and ATG-9 in the AIY interneurons of *C. elegans*. AIYs are a pair of bilaterally symmetric interneurons that display a stereotyped distribution of presynaptic specializations along their neurites (Fig 1A and 1C) [30,31]. Simultaneous visualization of ATG-9::GFP and the presynaptic marker mCherry::RAB-3 and Synaptogyrin (SNG-1)::BFP indicated that

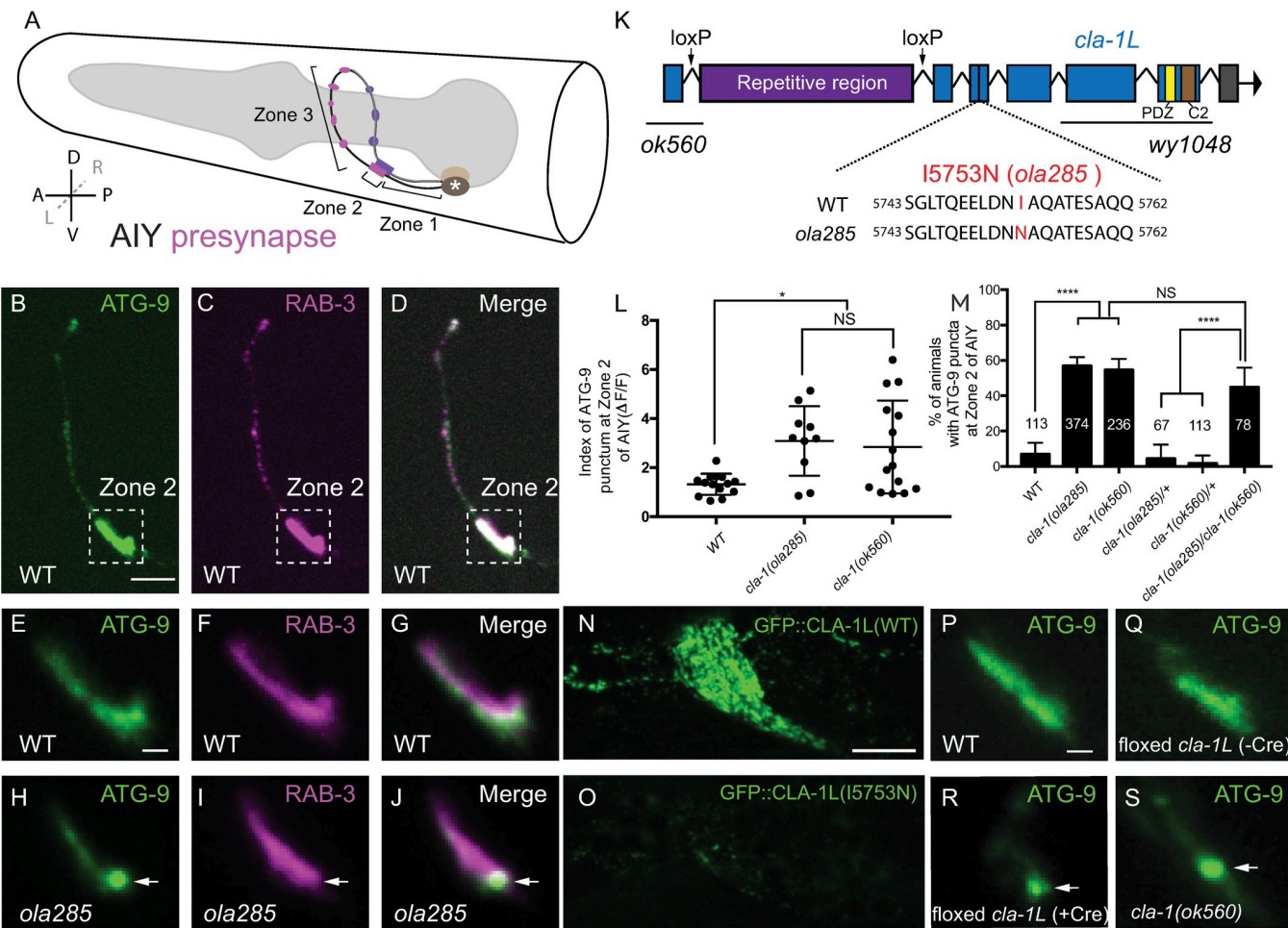

**Fig 1. The long isoform of Clarinet (CLA-1L) regulates ATG-9 trafficking at presynaptic sites.** (**A**) Schematic of the head of *C. elegans*, including pharynx (grey region) and the 2 bilaterally symmetric AIY interneurons. The asterisk denotes the cell body. There are 3 distinct segments along the AIY neurite: an asynaptic region proximal to AIY cell body (Zone 1), a large presynaptic region (Zone 2), and a segment with discrete presynaptic clusters at the distal part of the neurite (Zone 3) [30,31]. Presynaptic regions (Zone 2 and Zone 3) are in magenta (AIYL) or violet (AIYR). In axis, A, anterior; P, posterior; L, left; R, right; D, dorsal; V, ventral. (**B-D**) Distribution of ATG-9::GFP (**B**) and synaptic vesicle protein (mCherry::RAB-3, pseudo-colored magenta) (**C**) in the synaptic regions of AIY (merge in **D**). The dashed box encloses AIY Zone 2. (**E-J**) Distribution of ATG-9::GFP (**E** and **H**) and synaptic vesicle protein (mCherry::RAB-3, pseudo-colored magenta) (**F** and **I**) at Zone 2 of AIY (merge in **G** and **J**) in wild-type (WT) (**E-G**) and *ola285* mutant (**H-J**) animals. ATG-9 is evenly distributed in WT but forms subsynaptic foci in *ola285* mutants, which are not enriched with RAB-3 (indicated by arrows in **H-J**). (**K**) Schematic of the genomic region of *cla-1L*. The locations of loxP sites and the genetic lesions of the *cla-1* alleles examined in this study are indicated. The genetic lesion in allele *ola285* (I to N at residue 5753) is shown for both WT and *ola285* mutants. The positions of the repetitive region in CLA-1L and the conserved PDZ and C2 domains in all CLA-1 isoforms are also shown in the schematic. (**L**) Quantification of the index of ATG-9 punctum (ΔF/F; see Methods) at Zone 2 of AIY in wild-type (WT), *cla-1(ola285)*, and *cla-1(ok560)* mutants. Error bars show standard deviation (SD). "NS" (not significant), *$p < 0.05$ by ordinary one-way ANOVA with Tukey's multiple comparisons test. Each dot in the scatter plot represents a single animal. (**M**) Quantification of the percentage of animals displaying ATG-9 subsynaptic foci at AIY Zone 2 in the indicated genotypes. Error bars represent 95% confidence interval. "NS" (not significant), ****$p < 0.0001$ by two-tailed Fisher's exact test. The number on the bars indicates the number of animals scored. (**N, O**) Endogenous expression of GFP::CLA-1L (WT) (**N**) and GFP::CLA-1L (I5753N) (**O**) in the *C. elegans* nerve ring. (**P-S**) Distribution of ATG-9::GFP at Zone 2 of AIY in wild-type (WT) (**P**), *floxed cla-1L without Cre* (**Q**), and *floxed cla-1L with Cre* expressed cell specifically in AIY (**R**) and *cla-1(ok560)* (**S**) animals. Arrows (in **R** and **S**) indicate abnormal ATG-9 foci. Scale bar (in **B** for **B-D**), 5 μm; (in **E** for **E-J** and **P-S**), 1 μm; (in **N** for **N-O**), 10 μm. Data for Fig 1L and 1M can be found in S1 Data.

ATG-9 is enriched at presynaptic sites in AIY, consistent with previous studies (Figs 1B–1G and 2A–2D) [16,24].

To identify molecular mechanisms that selectively disrupt ATG-9 sorting at synapses, we performed unbiased forward genetic screens. From our screens, we isolated several alleles that affected ATG-9 expression levels, trafficking to synapses or sorting at synaptic sites (S1 Table). We focused our studies on allele *ola285* for 2 reasons: (1) We had previously demonstrated

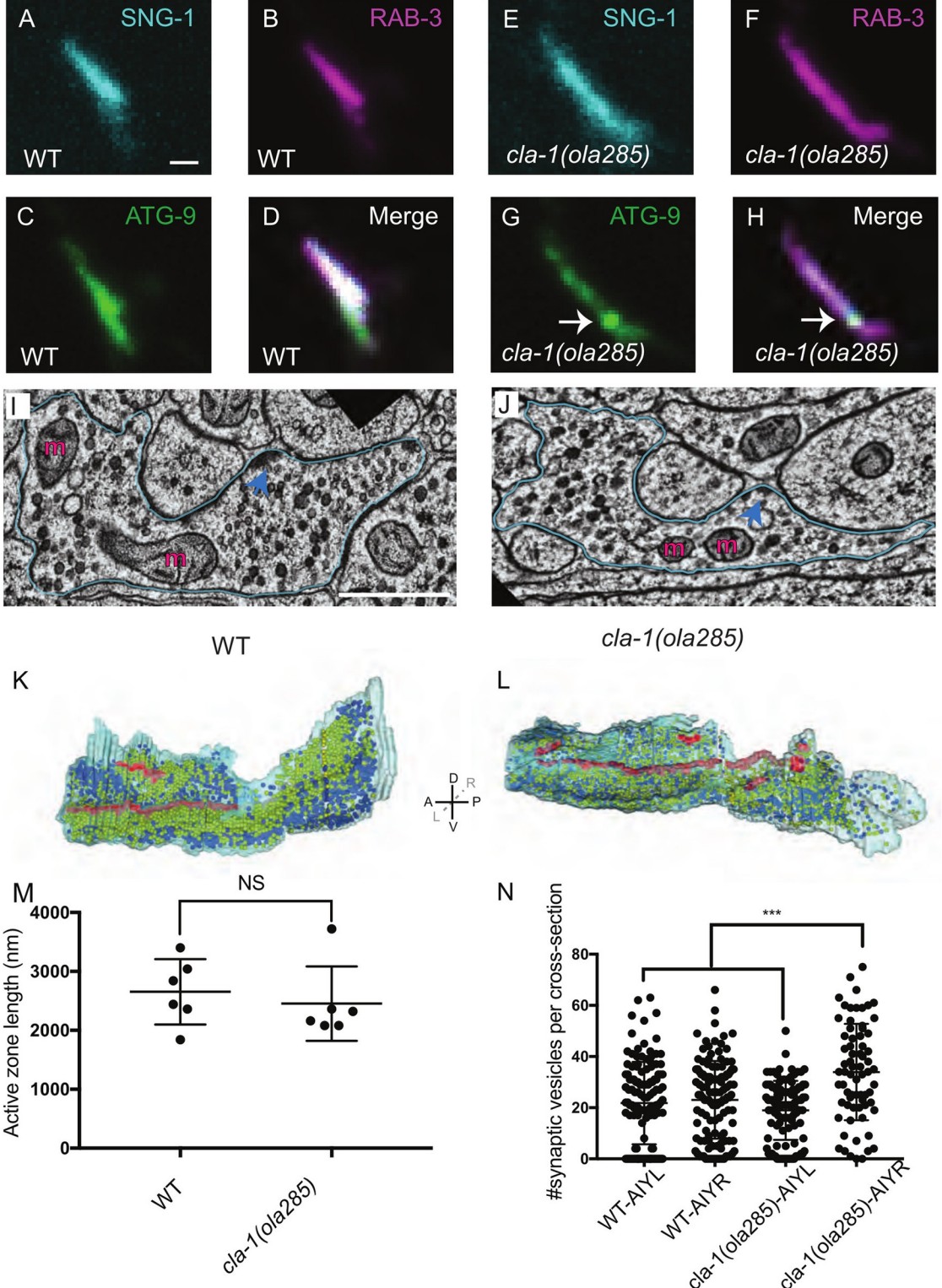

**Fig 2. ATG-9 and synaptic vesicle proteins are differentially regulated by CLA-1L.** (A-H) Distribution of SNG-1::BFP (pseudo-colored cyan) (**A** and **E**), mCherry::RAB-3 (pseudo-colored magenta) (**B** and **F**), and ATG-9::GFP (**C** and **G**) at Zone 2 of AIY (merge in **D** and **H**) in wild-type (WT) (**A-D**) and *cla-1(ola285)* mutant (**E-H**) animals. While we observe a phenotype for abnormal ATG-9 distribution to subsynaptic foci in *cla-1(ola285)* mutants (indicated by arrows in **G** and **H**), we do not see a similar redistribution for synaptic vesicle proteins SNG-1 and RAB-3. (**I, J**) Electron microscopy of the Zone 2 region in wild-type (**I**) and *cla-1(ola285)* mutant

animals (**J**). Blue lines, outline of AIY Zone 2. Presynaptic dense projections, pointed with arrows in dark blue. "m", mitochondria. (**K**, **L**) Electron micrograph reconstructions of AIY Zone 2 in wild-type (**K**) and *cla-1(ola285)* mutant animals (**L**). The active zones (or dense projections) are highlighted in red. Synaptic vesicles and dense core vesicles are symbolized by yellow and blue spheres, respectively. In axis: A, anterior; P, posterior; L, left; R, right; D, dorsal; V, ventral. (**M**) Measurement of the length of the active zone (highlighted in red in **K** and **L**) in the AIY neurons of 3 wild-type and 3 *cla-1(ola285)* mutants. Error bars represent standard deviation (SD). "NS" (not significant) by two-tailed Fisher's exact test. Each dot in the scatter plot represents a single neuron. (**N**) Quantification of synaptic vesicles in the AIY neurons (AIY-L: AIY on the left side; AIY-R: AIY on the right side) of 1 wild-type and 1 *cla-1(ola285)* mutant. The differences in AIYR (observed in this figure for *cla-1(ola285)* mutants, also in S3 for endosomal area in wild type) are consistent with previous findings about asymmetry of AIY neurons [31], including gene expression [114]. Nonetheless, unlike for other examined genotypes by EM (like UNC-26/Synaptojanin) [24], the observed phenotypes do not reveal major differences that could account for the observed light microscopy phenotypes of ATG-9. Error bars represent standard deviation (SD). ***$p < 0.001$ by ordinary one-way ANOVA with Tukey's multiple comparisons test. Each dot in the scatter plot represents a single section. Scale bar (in **A** for **A-H**), 1 μm; (in **I** for **I** and **J**), 500 nm. Data for Fig 2M and 2N can be found in S1 Data.

that Synaptojanin 1 is necessary for ATG-9 sorting at presynaptic sites [24], and *ola285* allele phenocopies the defects observed for *unc-26/synaptojanin 1* mutants (namely, that ATG-9 abnormally localizes to subsynaptic foci); (2) in *ola285* allele, the localization of ATG-9 at synapses is differentially affected as compared to the localization of synaptic vesicle proteins, suggesting that *ola285* allele specifically affects ATG-9 sorting at synapses.

In wild-type animals, ATG-9 is evenly distributed in the presynaptic-rich region (termed Zone 2; Fig 1E–1G and 1M), whereas in the *ola285* mutants, ATG-9 is enriched in subsynaptic foci in about 60% of animals (Fig 1H–1J and 1M). Unlike the ATG-9::GFP foci observed in *ola285* mutants, the synaptic vesicle markers SNG-1 (and RAB-3) retained their wild-type phenotype in *ola285* mutants in the Zone 2 region of AIY (Figs 1E–1J and 2A–2H). To quantify the expressivity of the phenotype, we calculated an index ($F_{peak}$-$F_{trough}$)/$F_{trough}$) for ATG-9:: GFP. The index was calculated from representative micrographs of the ATG-9 phenotype in the indicated genotypes (see Methods). Our quantifications of expressivity revealed a significant redistribution of ATG-9 to subsynaptic foci in *ola285* mutants as compared to wild-type animals (Fig 1L).

To identify the genetic lesion corresponding to the *ola285* allele, we performed single-nucleotide polymorphism (SNP) mapping coupled with whole-genome sequencing (WGS) [24,32–34]. We identified the genetic lesion of *ola285* in the locus of the gene *cla-1*, which encodes for Clarinet. Clarinet (CLA-1) is an active zone protein that contains PDZ and C2 domains with similarity to vertebrate active zone proteins Piccolo and RIM (Fig 1K; [25–27]). Three lines of evidence support that *ola285* is an allele of *cla-1*. First, *ola285* contains a missense mutation in the *cla-1* gene that converts Isoleucine (I) to Asparagine (N) at the residue 5753 (I5753N) (Fig 1K). Second, an independent allele of *clarinet*, *cla-1(ok560)*, phenocopied the ATG-9 localization defects observed for *ola285* mutants, both in terms of penetrance and expressivity (Fig 1L and 1M). Third, transheterozygous animals carrying both alleles *ola285* and *cla-1(ok560)* resulted in abnormal ATG-9 localization at synapses, similar to that seen for either *ola285* or *cla-1(ok560)* homozygous mutants (Fig 1M). The inability of *cla-1(ok560)* to complement the newly isolated allele *ola285* supports that they correspond to genetic lesions within the same gene, *cla-1*. Together, our data indicate that CLA-1, an active zone protein with similarity to *Drosophila* active zone protein Fife, and vertebrate active zone proteins Piccolo and RIM [25], is required for sorting of the autophagy protein ATG-9 at presynaptic sites.

## Clarinet long isoform, CLA-1L, acts cell autonomously to selectively regulate ATG-9 sorting at presynaptic sites

The *cla-1* gene encodes 3 isoforms: CLA-1L (long), CLA-1M (medium), and CLA-1S (short) (S1A Fig). The long isoform, which contains a repetitive region predicted to be disordered (Fig

1K), is necessary for synaptic vesicle clustering, whereas the shorter isoforms are required for active zone assembly [25]. The isolated allele *cla-1(ola285)* (a missense mutation in the coding region of *cla-1L*), as well as the examined allele *cla-1(ok560)* (a deletion of the promoter and part of the coding region of *cla-1L*), only affect CLA-1L, but not CLA-1M or CLA-1S. A null allele affecting all isoforms, *cla-1(wy1048)*, did not display a more severe ATG-9 phenotype than the alleles affecting only CLA-1L (S1A and S1B Fig). These findings suggest that the long isoform of Clarinet (CLA-1L) is necessary for presynaptic sorting of ATG-9.

To better assess the effects of the lesion of allele *ola285* in CLA-1 protein product in vivo, we inserted, via CRISPR, a DNA sequence encoding GFP at the 5′-end of the endogenous *cla-1* locus. The 5′-end of *cla-1* gene corresponds to the N-terminus of CLA-1L (S1A Fig), so the inserted GFP specifically labels CLA-1L (S6B Fig). We observed that CLA-1L expression levels were reduced in *ola285* as compared to wild-type animals (Figs 1N–1O and S1C). Based on the loss of function phenotype of *ola285* allele, we hypothesize that the missense mutation results in degradation of CLA-1L.

To determine the specific requirement of CLA-1L for ATG-9 sorting at presynaptic sites in AIY, we manipulated the expression of CLA-1L in AIY using a cell-specific knockout strategy [25]. We used a strain in which loxP sites were inserted, via CRISPR, to flank the unique 5′-end gene locus specific to CLA-1L (Figs 1K and S1A). Cell-specific expression of Cre recombinase in AIY, which leads to AIY-specific deletion of the CLA-1L isoform (without affecting CLA-1S and CLA-1M), resulted in the ATG-9 phenotype in AIY (Figs 1R and S1D), which was indistinguishable from that seen for the *cla-1 (ok560)* allele (Figs 1S and S1D, compare to wild type in Figs 1P, 1Q, and S1D). Together, our data indicate that the allele *ola285* affects the long isoform of Clarinet (CLA-1L) and that CLA-1L regulates presynaptic sorting of ATG-9 in a cell-autonomous manner.

## ATG-9-containing vesicles cluster at clathrin-rich subsynaptic domains in *cla-1(ola285)* mutants

In *cla-1(ola285)* animals, ATG-9 sorting defects phenocopied those observed for *unc-26/synaptojanin 1* mutant [24]. However, unlike *unc-26/synaptojanin 1*, *cla-1(ola285)* did not exhibit a mutant phenotype for synaptic vesicle proteins in presynaptic-rich Zone 2 (Figs 1E–1J, 2A–2H, and S2A). To better understand the effects of the genetic lesion of *cla-1(ola285)* on synaptic morphology and synaptic vesicle distribution, we performed transmission electron microscopy (EM) studies. Our ultrastructural analyses in the AIY Zone 2 region revealed that the average length of the active zone is similar between wild-type (2.65 ± 0.23 μm) and *cla-1 (ola285)* mutants (2.45 ± 0.26 μm) (examined in 6 EM reconstructed neurons per genotype; Figs 2I–2M, S3D, and S3E), consistent with fluorescence microscopy observations that CLA-1L is not required for active zone assembly (S3A–S3C Fig; [25]) and suggestive that the defects observed for ATG-9 missorting are not due to general defects in the AIY active zone. Quantification of the presence of synaptic vesicles, dense core vesicles, and endosomal structures in the electron micrographs of *cla-1(ola285)* and wild-type animals did not reveal major differences that could account for the observed phenotypes of ATG-9 (Figs 2I–2N and S3F-S3H). These observations are consistent with our light microscopy studies on the distribution of synaptic vesicle proteins in *cla-1(ola285)* mutants (Figs 1E–1G and 2A–2H) and stand in contrast to previous observations in *unc-26/synaptojanin 1* mutants, where a general defect in all vesicular structures is observed [24,35]. These findings suggest that the observed phenotype for ATG-9 in *cla-1(ola285)* mutants is not due to a general problem in synaptic morphology or synaptic vesicle endocytosis.

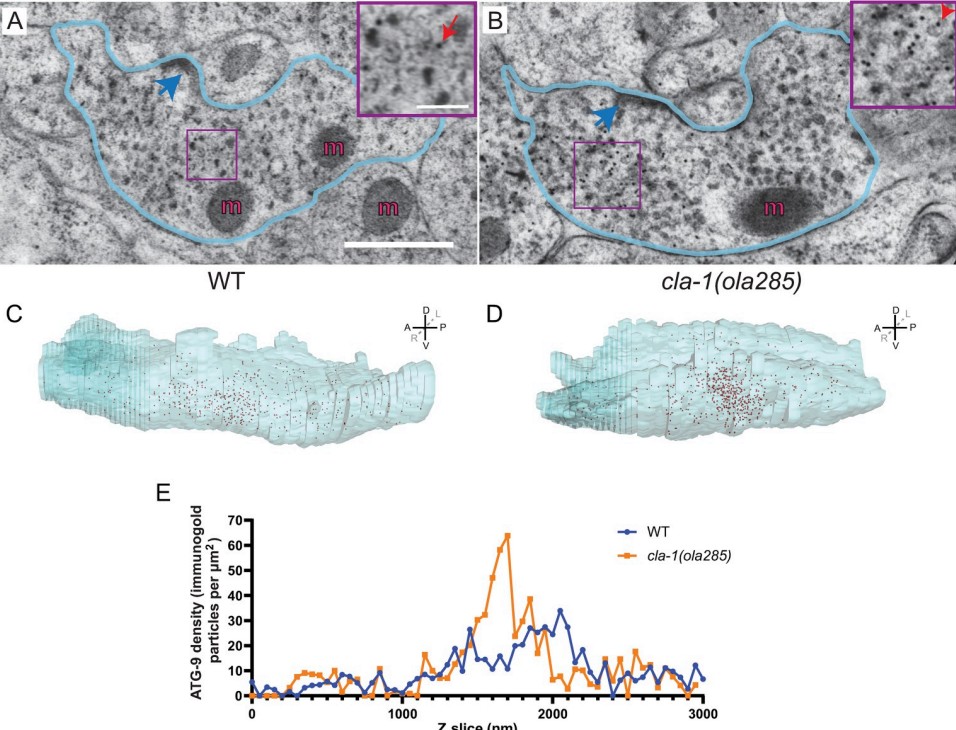

**Fig 3. ATG-9-containing vesicles cluster at subsynaptic domains in *cla-1(ola285)* mutants.** (**A**, **B**) Immunogold electron microscopy at Zone 2 of AIY neurons in wild-type (**A**) and *cla-1(ola285)* mutant (**B**) transgenic animals, with ATG-9::GFP panneuronally expressed and antibodies directed against GFP [24]. Blue line outlines the AIY Zone 2 region; dark blue arrows point at presynaptic dense projections. "m", mitochondria. Insets at the upper right hand corner correspond to higher magnifications of the regions highlighted with purple squares, with red arrows pointing to a representative immunogold particle detecting ATG-9::GFP in vesicular structures. (**C**, **D**) Electron micrograph reconstructions of Zone 2 of AIY and ATG-9::GFP immunogold particles in wild-type (**C**) and *cla-1(ola285)* mutant animals (**D**). Red dots: ATG-9::GFP immunogold particles. In axis: A, anterior; P, posterior; L, left; R, right; D, dorsal; V, ventral. (**E**) Distribution of ATG-9 immunogold particles density per cross-section in wild-type (blue line and round dots) and *cla-1(ola285)* mutant animals (orange line and square dots). X axis, Z slices at Zone 2 along the antero-posterior axis. Scale bar (in **A** for **A** and **B**), 500 nm; (in insert of **A** for inserts of **A** and **B**), 100 nm. Data for Fig 3E can be found in S1 Data.

To better understand the distribution of ATG-9 in *cla-1(ola285)* mutants, we performed immuno-EM studies and stained ATG-9::GFP. Although the total number of ATG-9::GFP gold particles displays no significant difference between wild-type and *cla-1(ola285)* mutant animals (number of gold particles = 503 for wild-type and 599 for *cla-1* mutants), the particle distribution displayed differences. In wild-type animals, ATG-9::GFP gold particles are distributed along the Zone 2 synapse (Fig 3A, 3C, and 3E). In *cla-1(ola285)* mutants, however, we observe ATG-9::GFP gold particles concentrate on subsynaptic regions (Fig 3B, 3D, and 3E). These findings are consistent with our fluorescence microscopy data that ATG-9 localizes to a subsynaptic region in *cla-1(ola285)* mutants (Fig 1H). Together, our findings indicate that the ATG-9 phenotype in *cla-1(ola285)* results from differences in the distribution of ATG-9-containing vesicular structures at the synapse.

To determine if the abnormal accumulation of ATG-9 in *cla-1(ola285)* mutants, like that observed for *unc-26/synaptojanin 1*, results from defects in ATG-9 sorting during exo-endocytosis, we next examined the necessity of synaptic vesicle exocytosis proteins in the ATG-9 phenotype of *cla-1(ola285)* mutants. We first visualized ATG-9 in putative null alleles *unc-13(s69)/Munc13*, *unc-10(md1117)/RIM*, *unc-18(e81)/Munc18*, and *unc-2(e55)/CaV2α1* (voltage-gated

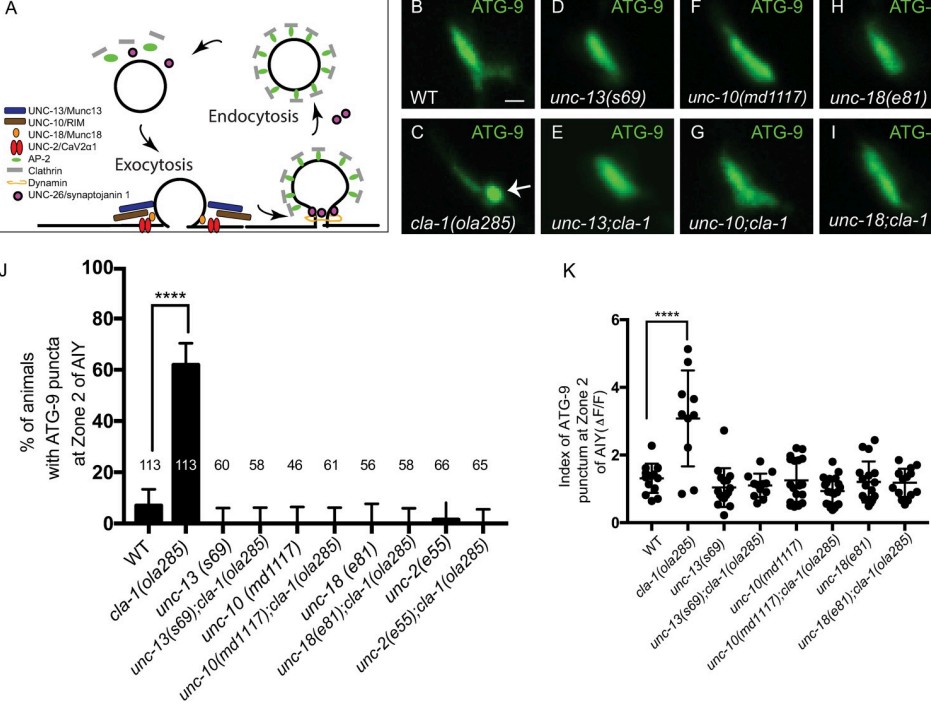

**Fig 4. ATG-9 foci in *cla-1(ola285)* mutants are suppressed by mutants for synaptic vesicle exocytosis. (A)**
Schematic of the proteins required for the synaptic vesicle cycle and associated with this study (both the names used
for *C. elegans* and vertebrates are listed). (**B-I**) Distribution of ATG-9::GFP at Zone 2 of AIY in wild-type (WT) (**B**),
*cla-1(ola285)* (**C**), *unc-13(s69)* (**D**), *unc-13(s69);cla-1(ola285)* (**E**), *unc-10 (md1117)* (**F**), *unc-10(md1117);cla-1(ola285)*
(**G**), *unc-18(e81)* (**H**), and *unc-18(e81);cla-1(ola285)* (**I**) animals. ATG-9 subsynaptic foci are indicated by the arrow (in
**C**). (**J**) Quantification of the percentage of animals displaying ATG-9 subsynaptic foci at AIY Zone 2 in the indicated
genotypes. The data used to quantify the percentage of animals displaying ATG-9 subsynaptic foci in wild type are the
same as those in Fig 1M (explained in Methods). Error bars represent 95% confidence interval. ****$p < 0.0001$ by two-
tailed Fisher's exact test. The number on the bars indicates the number of animals scored. (**K**) Quantification of the
index of ATG-9 punctum (ΔF/F) at Zone 2 of AIY in the indicated genotypes. The data used to quantify the index of
ATG-9 punctum (ΔF/F) in wild-type and *cla-1(ola285)* mutants are the same as those in Fig 1L (explained in
Methods). Error bars show standard deviation (SD). ****$p < 0.0001$ by ordinary one-way ANOVA with Tukey's
multiple comparisons test. Each dot in the scatter plot represents a single animal. Scale bar (in **B** for **B-I** and **J-Q**),
1 μm. Data for Fig 4J and 4K can be found in S1 Data.

calcium channels), all of which encode proteins essential for synaptic vesicle fusion (Fig 4A)
[36–43]. Single mutants of *unc-13(s69)*, *unc-10(md1117)*, *unc-18(e81)/Munc18*, and *unc-2(e55)*
did not disrupt ATG-9 localization (Fig 4D, 4F, 4H, 4J, and 4K). Double mutants of *unc-13
(s69);cla-1(ola285)*, *unc-10(md1117);cla-1(ola285)*, *unc-18(e81);cla-1(ola285)*, and *unc-2(e55);
cla-1(ola285)* completely suppressed abnormal ATG-9 localization in *cla-1* mutants (Fig 4E,
4G, 4I, 4J, and 4K). These results are consistent with previous findings that ATG-9-positive
vesicles undergo exo-endocytosis at presynaptic sites by using the synaptic vesicle cycling
machinery [24] and suggest that the ATG-9 phenotype in *cla-1(ola285)* mutants results from
defects in ATG-9 sorting upon ATG-9 exo-endocytosis.

In *unc-26/synaptojanin 1* mutants, ATG-9 abnormally colocalizes with the clathrin heavy
chain subunit, CHC-1 [24]. To determine if, in *cla-1(ola285)* mutants, ATG-9 similarly abnor-
mally colocalizes with the clathrin heavy chain subunit, we examined the colocalization
between ATG-9 and CHC-1 in wild-type and *cla-1(ola285)* mutant animals (Fig 5A–5H). We
observed that both ATG-9 and CHC-1 abnormally localized to similar synaptic foci in *cla-1
(ola285)* mutants (Figs 5E–5H and S2B). Together, our findings suggest that in *cla-1* mutants,
ATG-9-containing vesicles abnormally cluster at clathrin-rich subsynaptic domains.

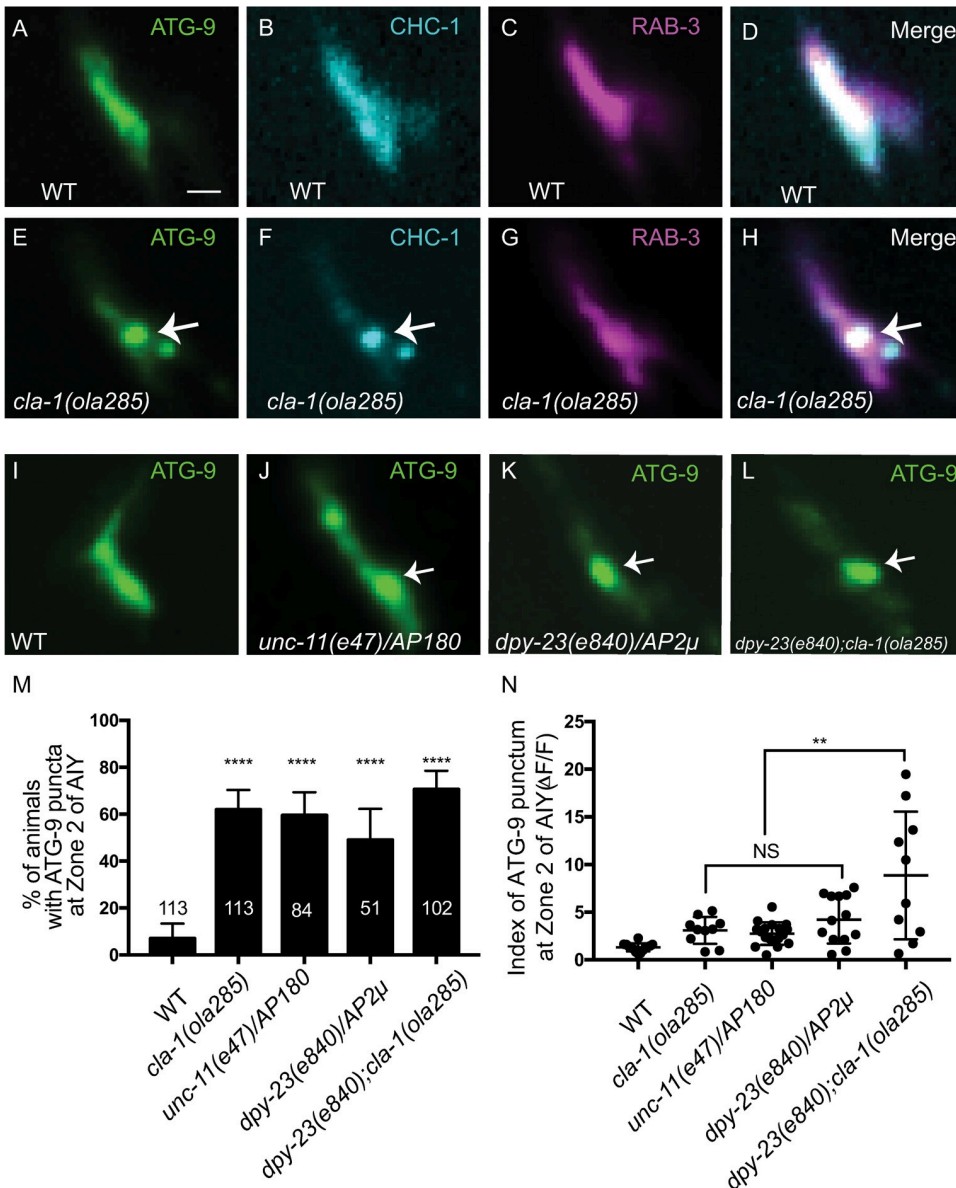

**Fig 5. The clathrin-associated adaptor complexes, AP-2 and AP180, regulate ATG-9 trafficking at presynaptic sites.** (**A**-**D**) Distribution of ATG-9::GFP (**C**), BFP::CHC-1 (pseudo-colored cyan) (**D**), and mCherry::RAB-3 (pseudo-colored magenta) (**E**) at Zone 2 of AIY (merge in **F**) in wild-type (WT) animals. (**E**-**H**) Distribution of ATG-9::GFP (**A**), BFP::CHC-1 (pseudo-colored cyan) (**B**), and mCherry::RAB-3 (pseudo-colored magenta) (**C**) at Zone 2 of AIY (merge in **D**) in *cla-1(ola285)* mutant animals. ATG-9 subsynaptic foci are enriched with CHC-1 in *cla-1(ola285)* mutants (indicated by arrows in **A**, **B**, and **D**). (**I**-**L**) Distribution of ATG-9::GFP at Zone 2 of AIY in wild-type (WT) (**E**), *unc-11(e47)/AP180* (**F**), *dpy-23(e840)/AP2μ* (**G**), and *dpy-23(e840);cla-1(ola285)* (**H**) mutant animals. Abnormal ATG-9 subsynaptic foci are indicated by arrows in **F**-**H**. (**M**) Quantification of the percentage of animals displaying ATG-9 subsynaptic foci at AIY Zone 2 for the indicated genotypes. The data used to quantify the percentage of animals displaying ATG-9 subsynaptic foci in wild-type and *cla-1(ola285)* mutants are the same as those in Fig 4J (explained in Methods). Error bars represent 95% confidence interval. ****$p < 0.0001$ by two-tailed Fisher's exact test. The number on the bars indicates the number of animals scored. (**N**) Quantification of the index of ATG-9 punctum ($\Delta F/F$) at Zone 2 of AIY for the indicated genotypes. The data used to quantify the index of ATG-9 punctum ($\Delta F/F$) in wild-type and *cla-1(ola285)* mutants are the same as those in Fig 1L (explained in Methods). Error bars show standard deviation (SD). "NS" (not significant), **$p < 0.01$ by ordinary one-way ANOVA with Tukey's multiple comparisons test. Each dot in the scatter plot represents a single animal. Scale bar (in **A** for **A**-**L**), 1 μm. Data for Fig 5M and 5N can be found in S1 Data.

## The clathrin adaptor complexes, AP-2 and AP180, regulate ATG-9 sorting at presynaptic sites

We next examined the genetic relationship between clathrin adaptor protein complexes and CLA-1L in ATG-9 localization. The AP-2 complex mediates clathrin-mediated endocytosis (CME) of synaptic vesicle proteins [44–49], and it has been implicated in the sorting of ATG-9 during autophagy induction in mammalian nonneuronal cells [50–52]. To determine if the AP-2, and the associated AP180, adaptor complexes were required in presynaptic sorting of ATG-9, we examined ATG-9 localization in the null alleles *dpy-23(e840)/AP2µ* and *unc-11 (e47)/AP180*. We observed that *dpy-23(e840)/AP2µ* and *unc-11(e47)/AP180* mutants phenocopied *cla-1(ola285)* mutants in ATG-9 presynaptic sorting defects (Fig 5J, 5K, 5M, and 5N). In addition, the expressivity of the ATG-9 sorting defects was enhanced in *dpy-23(e840)/ AP2µ;cla-1(ola285)* double mutant worms (Fig 5L–5N). These findings suggest shared mechanisms that similarly result in defective ATG-9 sorting when clathrin-associated adaptor complexes, or the active zone protein CLA-1L, are disrupted.

## ATG-9 is sorted to endocytic intermediates via SDPN-1/syndapin 1 and the AP-1 adaptor complex

During endocytosis, clathrin-associated adaptor complexes mediate internalization and sorting of cargoes from both the plasma membrane and intracellular endocytic intermediates [53–55]. Since ATG-9 abnormally localizes to subsynaptic foci when disrupting the AP-2 (or the associated AP180) adaptor complexes, we reasoned that the subsynaptic ATG-9-rich foci might represent endocytic intermediates, from which the AP-2 adaptor complex binds to and sort out cargoes. We then sought to identify upstream molecules that mediate the sorting of ATG-9 to the endocytic intermediates. Disrupting those molecules should suppress ATG-9 foci in mutants for CLA-1L or AP-2.

We first examined SDPN-1/syndapin 1, a protein known to play important roles in early stages of membrane invagination during both activity-dependent bulk endocytosis (ADBE) [56,57] and ultrafast endocytosis of synaptic vesicles [58]. We reasoned that if ATG-9-containing vesicles were sorted via SDPN-1-dependent mechanisms, then *sdpn-1* mutants would suppress the observed ATG-9 foci for *cla-1(ola285)* and for mutants of the clathrin-associated adaptor complexes. Consistent with our hypothesis, we observed that ATG-9 localization was not disrupted in *sdpn-1(ok1667)* single mutants and that the abnormal ATG-9 foci were suppressed in *sdpn-1(ok1667);cla-1(ola285)* and *sdpn-1(ok1667);unc-11(e47)/AP180* double mutant animals (Fig 6A–6H). These findings are consistent with a requirement of SDPN-1 in the sorting of ATG-9 upstream of CLA-1L and clathrin-associated adaptor complexes.

Next, we examined the AP-1 adaptor complex, which acts at presynaptic sites to mediate endosomal sorting of ADBE [54,59,60]. To determine the requirement of the AP-1 adaptor complex in ATG-9 sorting at presynaptic sites, we examined ATG-9 localization in *unc-101 (m1)/AP1µ1* single mutant animals and in double mutants with *cla-1(ola285)*, *dpy-23(e840)/ AP2µ*, and *unc-11(e47)/AP180*. We observed that while *unc-101(m1)/AP1µ1* single mutant animals did not display detectable phenotypes in ATG-9 localization (Figs 7C, 7I and S4E), *unc-101(m1)/AP1µ1* suppressed the ATG-9 phenotype in double mutant animals (Figs 7A–7D, 7G–7I, and S4C-S4E). We confirmed this result by making double mutants of *cla-1(ola285)* with another allele, *unc-101(sy108)*, and observed it also suppressed the ATG-9 phenotype in *cla-1(ola285)* (S4E Fig). Furthermore, single-cell expression of the *C. elegans* cDNA of *unc-101 (m1)/AP1µ1* in AIY in the *unc-101(m1);cla-1(ola285)* double mutants reverted the phenotype, indicating that AP-1 acts cell autonomously in AIY to suppress the ATG-9 phenotype in *cla-1 (L)* (Figs 7E, 7I, and S4E). Our findings indicate that, similar to SDPN-1, the AP-1 adaptor

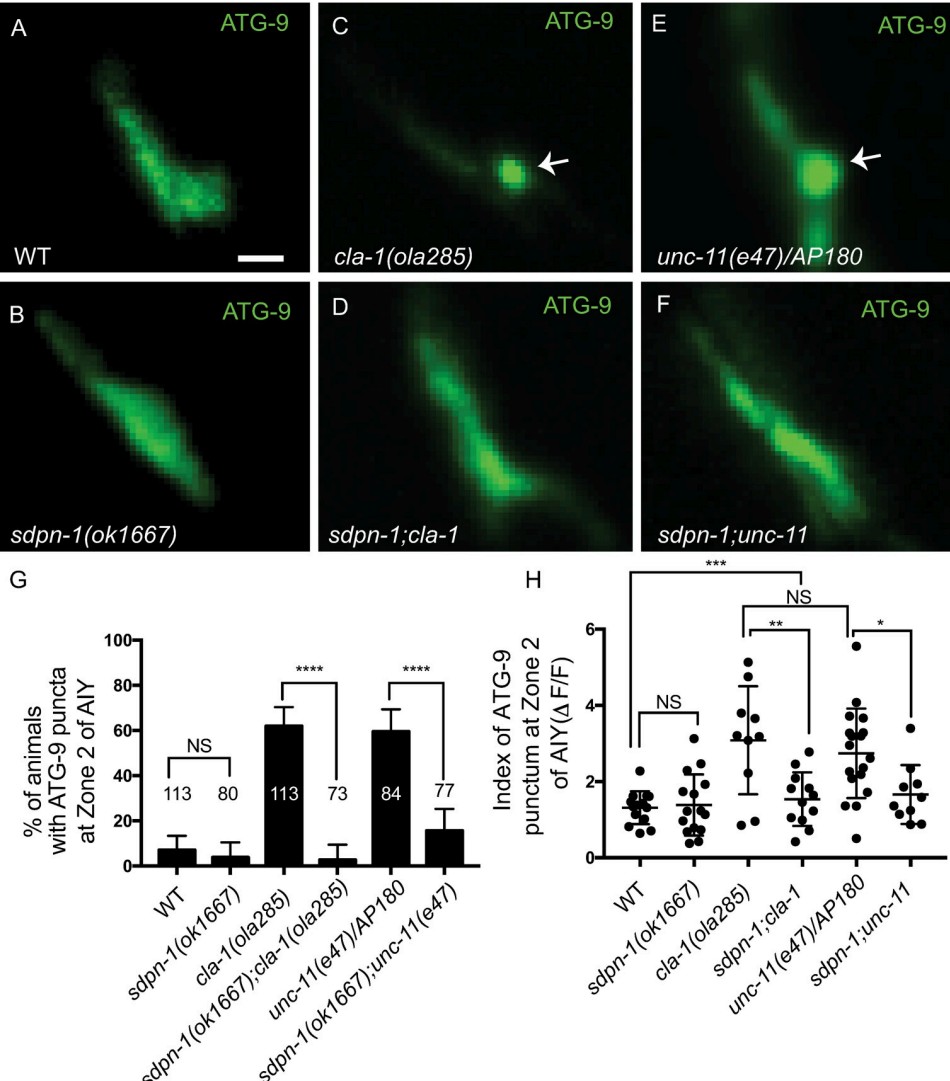

**Fig 6. SDPN-1/syndapin 1 regulates ATG-9 sorting at presynaptic sites.** (**A-F**) Distribution of ATG-9::GFP at Zone 2 of AIY in wild-type (WT) (**A**), *sdpn-1(ok1667)* (**B**), *cla-1(ola285)* (**C**), *sdpn-1(ok1667);cla-1(ola285)* (**D**), *unc-11(e47)/ AP180* (**E**), and *sdpn-1(ok1667);unc-11(e47)* (**F**) mutant animals. Abnormal ATG-9 subsynaptic foci are indicated by arrows in **C** and **E**. Note that mutations in SDPN-1/syndapin 1 suppress the abnormal ATG-9 phenotypes in *cla-1* and *unc-11/AP180* mutants. (**G**) Quantification of the percentage of animals displaying abnormal ATG-9 subsynaptic foci at AIY Zone 2 for the indicated genotypes. The data used to quantify the percentage of animals displaying ATG-9 subsynaptic foci in wild-type and *cla-1(ola285)* mutants are the same as those in Fig 4J; the data used in *unc-11(e47)* are the same as those in Fig 5M (explained in Methods). Error bars represent 95% confidence interval. "NS" (not significant), ****$p < 0.0001$ by two-tailed Fisher's exact test. The number on the bars indicates the number of animals scored. (**H**) Quantification of the index of ATG-9 punctum (ΔF/F) at Zone 2 of AIY for the indicated genotypes. The data used to quantify the index of ATG-9 punctum (ΔF/F) in wild-type and *cla-1(ola285)* mutants are the same as those in Fig 1L; the data used in *unc-11(e47)* are the same as those in Fig 5N (explained in Methods). Error bars show standard deviation (SD). "NS" (not significant), *$p < 0.05$, **$p < 0.01$, ***$p < 0.001$ by ordinary one-way ANOVA with Tukey's multiple comparisons test. Each dot in the scatter plot represents a single animal. Scale bar (in **A** for **A-F**), 1 μm. Data for Fig 6G and 6H can be found in S1 Data.

complex is required to sort ATG-9 at synapses, likely upstream of CLA-1L and clathrin-associated adaptor complexes.

The *C. elegans* UNC-101/AP1μ1 is more similar to the murine AP1μ1 (Query Cover: 100%; Percentage Identity: 74%) than to the murine AP2μ (Query Cover: 98%; Percentage Identity:

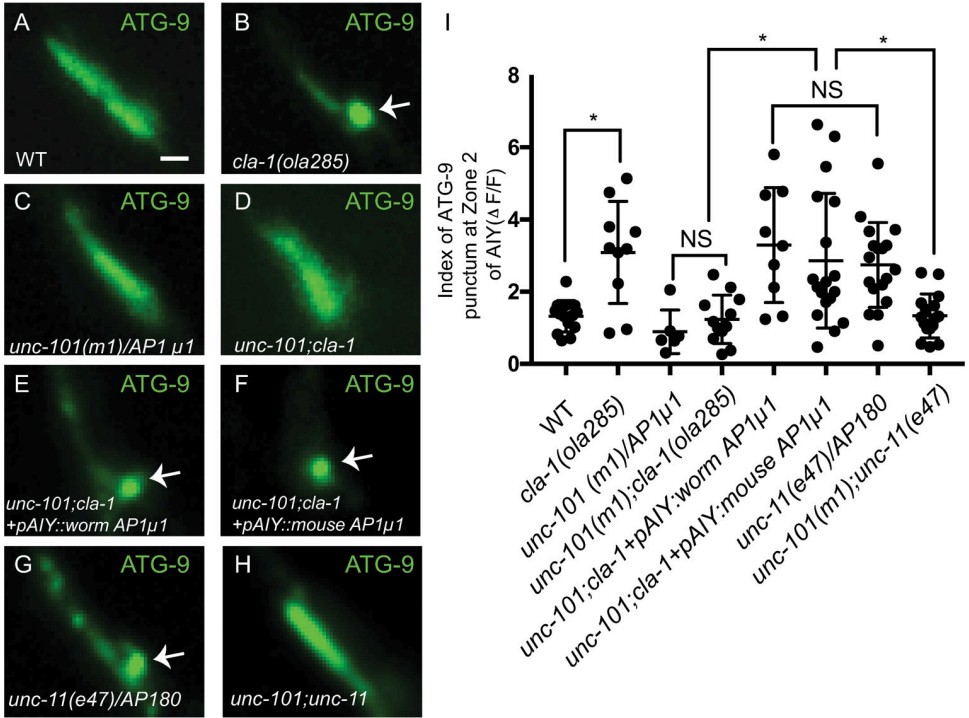

**Fig 7. ATG-9 is sorted to the endocytic intermediates via the AP-1 adaptor complex.** (A-H) Distribution of ATG-9::GFP at Zone 2 of AIY in wild-type (WT) (**A**), *cla-1(ola285)* (**B**), *unc-101(m1)/AP-1μ1* (**C**), *unc-101(m1);cla-1 (ola285)* (**D**), *unc-101;cla-1* mutants with *C. elegans* UNC-101/AP-1μ1 cDNA cell specifically expressed in AIY (**E**), *unc-101;cla-1* mutants with mouse AP1μ1 cDNA cell specifically expressed in AIY (**F**), *unc-11(e47)/AP180* (**G**), and *unc-101(m1);unc-11(e47)* (**H**). Abnormal ATG-9 subsynaptic foci are indicated by arrows in **B** and **E-G**. (**I**) Quantification of the index of ATG-9 punctum (ΔF/F) at Zone 2 of AIY for indicated genotypes. The data used to quantify the index of ATG-9 punctum (ΔF/F) in wild-type and *cla-1(ola285)* mutants are the same as those in Fig 1L; the data used in *unc-11(e47)* are the same as those in Fig 5N (explained in Methods). Error bars show standard deviation (SD). "NS" (not significant), *$p < 0.05$ by ordinary one-way ANOVA with Tukey's multiple comparisons test. Each dot in the scatter plot represents a single animal. Scale bar (in **A** for **A-H**), 1 μm. Data for Fig 7I can be found in S1 Data.

41%) (S4F Fig). To examine the conserved role of UNC-101/AP1μ1 in ATG-9 sorting, we expressed murine cDNA of *AP1μ1* or *AP2μ* cell specifically in AIY of the *unc-101(m1);cla-1 (ola285)* double mutants and examined ATG-9 localization. We observed that expression of murine *AP1μ1*, but not *AP2μ* cDNA, reverted the suppression seen for *unc-101(m1);cla-1 (ola285)* double mutant animals (Figs 7F, 7I, S4B, and S4E), suggesting that this sorting function is conserved between murine AP1μ1 and *C. elegans* UNC-101/AP1μ1. Together, our findings are consistent with a model whereby ATG-9 is sorted to vesicular structures via synaptic machinery that includes CLA-1L, and also endosomal sorting proteins AP-1, SDPN-1, AP-2, and AP180 (S5 Fig).

## CLA-1L extends to the periactive zone and genetically interacts with endocytic proteins to regulate ATG-9 sorting

Next, we used cell biological approaches to understand the relationship between active zone protein CLA-1L and the endocytic sorting machinery, which localizes primarily to a different subsynaptic domain called the periactive zone [61–63]. Large active zone proteins that bear functional similarity to CLA-1L, such as Piccolo and Bassoon, extend from the active zone subdomains to the periactive zone, a property that has been hypothesized to be important in their

roles sorting synaptic components during exo-endocytosis [62,64–70]. CLA-1L is twice the size as Piccolo and Bassoon and contains largely disordered regions that could facilitate its extension from the active zone to the neighboring periactive zones. Consistent with this, in previous studies, we documented that while the C-terminus of CLA-1 localized specifically to the active zone, the unique N-terminus of CLA-1L isoform localized beyond the active zone subdomain [25].

We compared the endogenous C-terminally tagged CLA-1::GFP, or the endogenous N-terminally tagged GFP::CLA-1L (S6A and S6B) [25], with a periactive zone marker APT-4/APA-2/AP-2α [71]. While the C-terminally tagged CLA-1::GFP specifically localizes to small puncta corresponding to the active zone (Fig 8A and 8D), the N-terminally tagged GFP::CLA-1L displays a more distributed presynaptic pattern, extending to other regions of the synaptic bouton beyond the active zone (Fig 8H and 8K). Importantly, we observed that the N-terminally tagged GFP::CLA-1L, but not the C-terminally tagged CLA-1::GFP, colocalizes with the endocytic marker APT-4/APA-2/AP-2α at the periactive zones (Fig 8A–8O). These findings suggest that the long isoform of CLA-1 is anchored, via its C-terminus, to the active zone, but extends to the periactive zone where the endocytic sorting machinery is present.

Based on the localization of CLA-1L to these presynaptic subdomains, and the ATG-9 phenotypes observed in *cla-1* and endocytic mutants, we hypothesized the existence of genetic interactions between CLA-1L and endocytic proteins at the periactive zone. To test this hypothesis, we examined the localization of ATG-9 in the double mutants of *cla-1(ola285)* with genes encoding periactive zone endocytic proteins, *ehs-1(ok146)/EPS15* or *itsn-1(ok268)/ intersectin 1*. We focused our analyses on the endocytic regulators EHS-1/EPS15 and ITSN-1/ intersectin 1 because of their hypothesized roles in coupling synaptic vesicle exocytosis at the active zone, and endocytosis at the periactive zone [61,62,72–79]. We observed that in null alleles of *ehs-1(ok146)/EPS15* and *itsn-1(ok268)/intersectin 1*, 30% of worms displayed abnormal ATG-9 foci (compared to 60% in *cla-1(L)* mutants) (Figs 8P, S7C, and S7E). Interestingly, we observed that *ehs-1(ok146);cla-1(ola285)* and *itsn-1(ok268);cla-1(ola285)* enhanced the ATG-9 phenotype as compared to any of the single mutants (S7A–S7G Fig), both in terms of penetrance (Fig 8P) and expressivity (S7G Fig). Our findings uncover a cooperative genetic relationship between CLA-1L and the EHS-1-ITSN-1 endocytic scaffolding complex, suggesting that the active zone protein CLA-1L acts in pathways that are partially redundant to the EHS-1-ITSN-1 complex in linking the active zone and periactive zone regions to regulate ATG-9 sorting at presynapses.

## Disrupted ATG-9 sorting in *cla-1(ola285)* mutants is associated with a deficit in activity-induced autophagosome formation

ATG-9 is important for autophagosome biogenesis at presynaptic sites [16,24]. To examine how autophagosome formation is affected in *cla-1(L)* mutants, we measured the average number of LGG-1/Atg8/GABARAP puncta (an autophagosomal marker) in the AIY neurites [12,16,24] (Fig 9A–9C). Previously, we observed the average number of LGG-1 puncta increased in AIY when the wild-type animals were cultivated at 25°C [12,24], a condition known to increase the activity state of the AIY neurons [80]. Worms with impaired exocytosis (in *unc-13* mutants) or impaired autophagy (in *atg-9* mutants) failed to display increased LGG-1 synaptic puncta [12,24]. We found that, unlike wild-type animals, the average number of LGG-1 puncta did not increase in *cla-1(L)* mutants (alleles *ola285* and *ok560*) in response to cultivation temperatures that increase the activity state of the neuron (Figs 9D and S8A). These findings indicate that activity-induced autophagosome formation at synapses is impaired in *cla-1(L)* mutants.

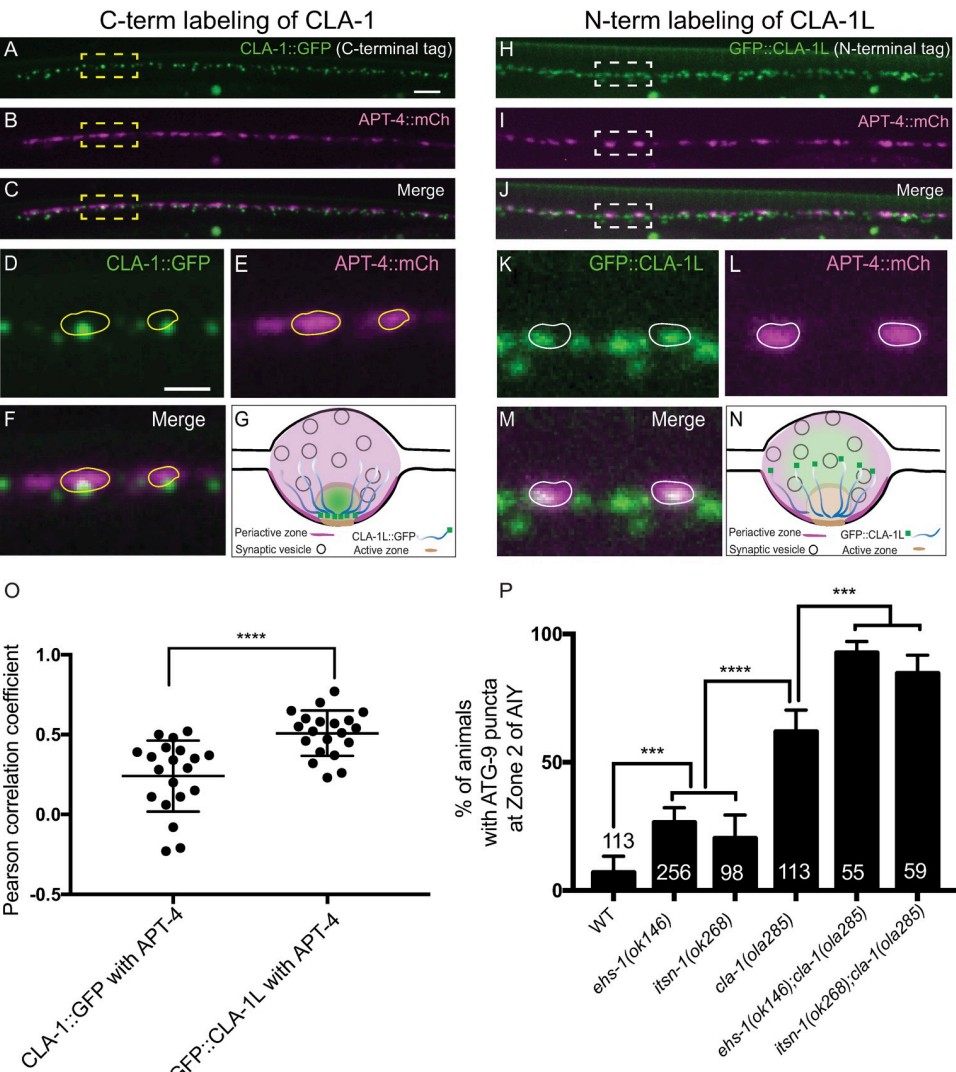

**Fig 8. CLA-1L genetically interacts with endocytic proteins at the periactive zone to regulate ATG-9 trafficking.**
(**A**-**C**) Distribution of endogenous C-terminally tagged CLA-1::GFP (**A**) and the endocytic zone marker APT-4/APA-2/AP-2α::mCherry (APT-4::mCh, pseudo-colored magenta) (**B**) in the neurons of the posterior dorsal nerve cord (merge in **C**) in wild-type animals. Note that APT-4::mCh is expressed in a subset of neurons in the dorsal nerve cord, driven by the punc-129 promoter, while CLA-1::GFP and GFP::CLA-1 are expressed panneuronally (so green puncta can be present where there are no magenta puncta; see Methods). (**D**-**F**) Enlarged regions enclosed in dashed boxes in **A**-**C**. Endogenous C-terminally tagged CLA-1::GFP (**D**) localizes to small puncta corresponding to the active zone [25], and different from the pattern observed for periactive zone protein, APT-4::mCh (**E**, merge in **F**). Yellow circles are drawn based on the outline of APT-4::mCh puncta in **E** and are located at the same positions in **D**-**F**. (**G**) Schematic of the localization of the C-terminally tagged CLA-1::GFP, relative to the subsynaptic active and periactive zones. (**H**-**J**) Distribution of endogenous N-terminally tagged GFP::CLA-1L (**H**) and the endocytic zone marker APT-4/APA-2/AP-2α::mCherry (APT-4::mCh, pseudo-colored magenta) (**I**) in neurons of the posterior dorsal nerve cord (merge in **J**) in wild-type animals. (**K**-**M**) Enlarged regions enclosed in dashed boxes in **H**-**J**. Endogenous N-terminally tagged GFP::CLA-1L (**K**) displays a more distributed synaptic distribution as compared to the C-terminally tagged CLA-1::GFP (compare with **A**, **D**, and **F**; see also [25]) and colocalizes with APT-4::mCh (**L**, merge in **M**). White circles are drawn based on the outline of APT-4::mCh puncta in **L** and are located at the same positions in **K**-**M**. (**N**) Schematic of the localization of the N-terminally tagged GFP::CLA-1L, relative to the subsynaptic active and periactive zones. (**O**) Pearson correlation coefficient for colocalization between CLA-1::GFP and APT-4::mCh, or between GFP::CLA-1L and APT-4::mCh, both in wild-type animals. Error bars show standard deviation (SD). ****$p < 0.0001$ by two-tailed unpaired Student $t$ test. Each dot in the scatter plot represents a single animal. (**P**) Quantification of the percentage of animals displaying ATG-9 subsynaptic foci at AIY Zone 2 in the indicated genotypes. The data used to quantify the percentage of animals displaying ATG-9 subsynaptic foci in wild-type and *cla-1(ola285)* mutants are the same as those in Fig 4J (explained in Methods). Error bars represent 95% confidence interval. ***$p < 0.001$,

****$p < 0.0001$ by two-tailed Fisher's exact test. The number on the bars indicates the number of animals scored. Scale bar (in **A** for **A**-**C** and **H**-**J**), 5 μm; (in **D** for **D**-**F** and **K**-**M**), 2 μm. Data for Fig 8O and 8P can be found in S1 Data.

Previously, we found that in mutants that affect early stages of autophagy such as *epg-9 (bp320)*, ATG-9 accumulated into subsynaptic foci, which colocalized with the clathrin heavy chain CHC-1 [24]. To determine a potential cross-talk between CLA-1L-mediated ATG-9 endocytosis and autophagy, we generated *epg-9(bp320);cla-1(ola285)* double mutant animals. The ATG-9 phenotype is enhanced in the double mutants, compared to single mutants (Fig 9E–9I). Our findings are consistent with a model whereby disrupted ATG-9 sorting in *cla-1(L)* mutants contributes to deficits in activity-induced autophagosome formation at synapses.

## Discussion

The active zone protein Clarinet (CLA-1L) regulates ATG-9 sorting at synapses and presynaptic autophagy. Autophagy, a conserved cellular degradative pathway, is temporally and

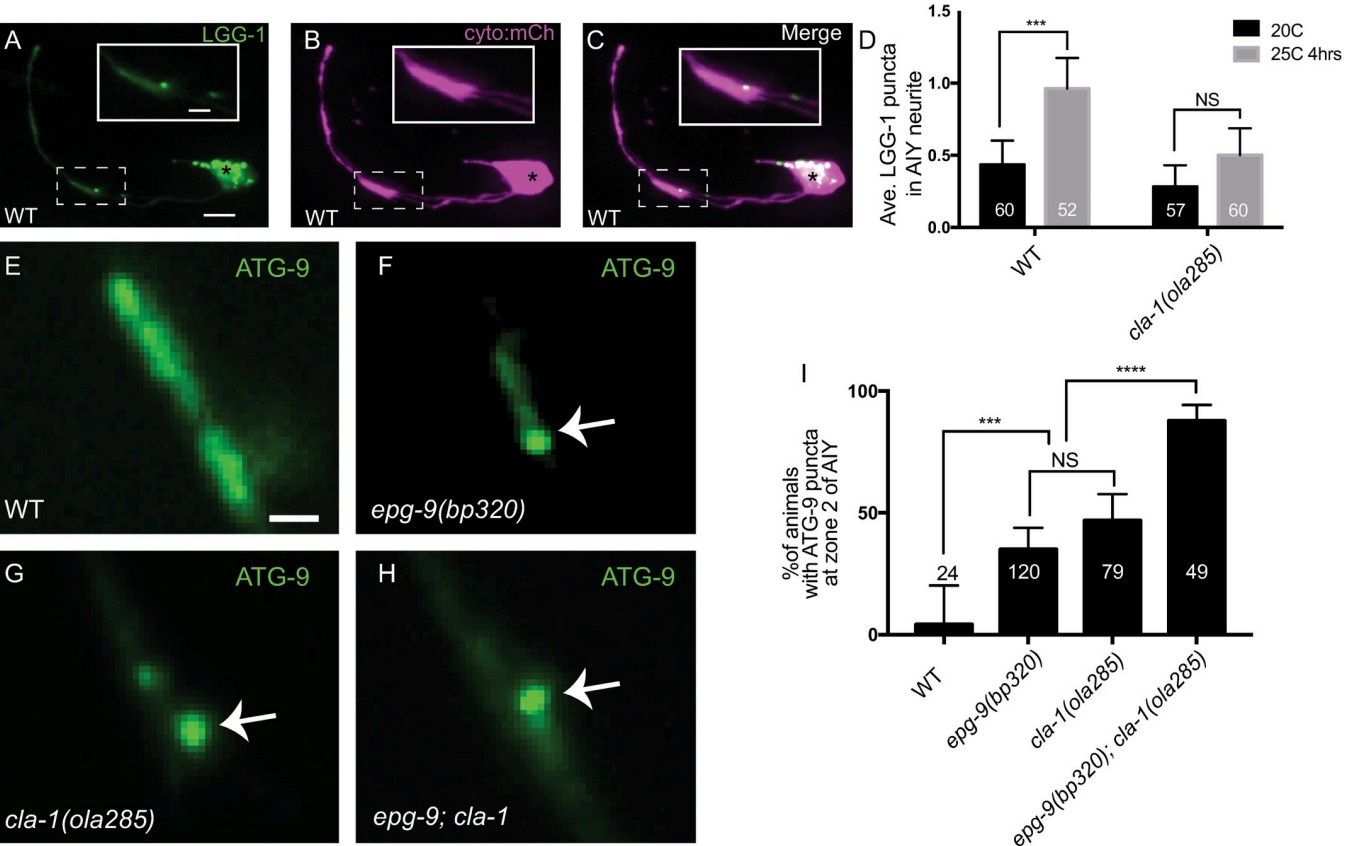

**Fig 9. Disrupted ATG-9 sorting in *cla-1(L)* mutants is associated with a deficit in activity-induced autophagosome formation.** (**A**-**C**) Confocal micrographs of GFP::LGG-1 (**A**) and cytoplasmic mCherry (cyto::mCh) (pseudo-colored magenta, **B**) in AIY (merge in **C**). Inset is the enlarged region enclosed in dashed box to show one LGG-1 punctum in AIY synaptic Zone 2. (**D**) Quantification of the average number of LGG-1 puncta in the AIY neurites at 20°C and at 25°C for 4 h in wild-type (WT) and *cla-1(ola285)* mutants. (Note: The activity state of the thermotaxis interneurons AIY is reported to increase when animals are cultivated at 25°C for 4 h, compared to animals at 20°C [80,115,116]. Error bars represent 95% confidence interval. "NS" (not significant), ***$p < 0.001$ by Kruskal–Wallis test with Dunn's multiple comparisons test. The number on the bars indicates the number of animals scored. (**E**-**H**) Distribution of ATG-9::GFP at Zone 2 of AIY in wild-type (**E**), *epg-9(bp320)* (**F**), *cla-1(ola285)* (**G**), and *epg-9(bp320); cla-1(ola285)* (**H**) mutant animals. Arrows (in **F**-**H**) indicate abnormal ATG-9 foci. (**I**) Quantification of the percentage of animals displaying ATG-9 subsynaptic foci at AIY Zone 2 in the indicated genotypes. Error bars represent 95% confidence interval. "NS" (not significant), ***$p < 0.001$, ****$p < 0.0001$ by two-tailed Fisher's exact test. The number on the bar indicates the number of animals scored. Scale bar (in **A** for **A**-**C**), 5 μm; (in inset of **A** for inset of **A**-**C**), 2 μm; (in **E** for **E**-**H**), 1 μm. Data for Fig 9D and 9I can be found in S1 Data.

spatially regulated in neurons to occur at synaptic compartments and in response to increased synaptic activity states [6,8–12,15,16,81,82]. How synaptic autophagy and synaptic activity states are coordinated in neurons is not well understood. In previous studies, we had determined that in *C. elegans* synapses and in vivo, ATG-9 is trafficked to synapses and that disruption of ATG-9 trafficking or its sorting results in disruptions to activity-dependent autophagosome biogenesis at synapses [16,24].

The synaptic machinery that sorts ATG-9 at presynaptic sites to regulate local autophagy remain largely unknown. In this study, by performing unbiased forward genetic screens, we uncover an unexpected role for the active zone protein Clarinet in synaptic sorting of ATG-9 and in activity-dependent autophagosome formation at synapses. We note that we did not detect differences in baseline autophagy in *cla-1* mutants, although we do observe defects in ATG-9 sorting under those conditions. Both ATG-9 sorting and autophagosome formation represent processes that are in flux, which increases upon neuronal activation. Our data suggest that the defects in ATG-9 sorting in *cla-1* mutants is not able to meet the needs for increased autophagy flux when neuronal activity increases.

The mechanisms of CLA-1L regulation of synaptic autophagy are likely distinct from those observed for other active zone proteins such as Bassoon. Previous studies in primary hippocampal neurons had demonstrated that active zone proteins such as Bassoon and Piccolo play important roles in synaptic protein homeostasis [83], in part via the regulation of presynaptic autophagy [84–86]. Bassoon negatively regulates presynaptic autophagy by serving as a scaffold for Atg5 [86], and the E3 ubiquitin ligase Parkin [84,85]. We find that instead of inhibiting autophagy, CLA-1L is required for activity-dependent synaptic autophagy, likely by sorting ATG-9 at synapses. Despite the mechanistic differences, together, these studies support the concept that in neurons, active zone proteins play important roles in regulating activity-dependent synaptic autophagy.

The CLA-1 long isoform, CLA-1L, extends from the active zone to the periactive zone and is specifically required for ATG-9 sorting by genetically interacting with proteins involved in endocytosis and sorting of synaptic cargo. Unlike the shorter isoforms, CLA-1L is not required for active zone assembly or synapse formation [25]. Our ultrastructural studies similarly demonstrate that morphological features of the synapse (including active zone length, synaptic vesicles, dense core vesicles, and endosomes) are largely unaffected in the *cla-1(ola285)* mutants, which specifically affect the CLA-1L isoform (Figs 2, S1G, and S1H). These findings suggest that the ATG-9 phenotype in *cla-1(L)* does not result from general defects in AIY synaptic morphology, or synaptic vesicle endocytosis. CLA-1L has been proposed to be functionally similar to Piccolo and Bassoon in its roles at the active zone [25]. In vertebrate synapses, Piccolo and Bassoon extend from the active zone region to periactive zones, and this architecture has been proposed to couple exocytosis at the active zone, and protein sorting during endocytosis at the periactive zone [62,64–70]. CLA-1L is an 8922 amino acid protein, twice the size of Bassoon (3942 amino acids) and Piccolo (4969 amino acids). It is anchored to the active zone via its C-terminus [25], with a large disordered N-terminal domain extending to the periactive zone, where endocytic processes occur. Consistent with these observations on CLA-1L size and position at the synapses, we uncover genetic interactions between CLA-1L and periactive zone proteins EHS-1/EPS15 or ITSN-1/intersectin 1, which have been suggested as linkers between active zone exocytosis and periactive zone endocytosis [61,62,72–79]. We propose that the specific requirement of CLA-1L in sorting ATG-9 at synapses is mediated via its capacity to extend across presynaptic subdomains, from the exocytic active zone to the endocytic periactive zone.

CLA-1L selectively regulates the sorting of ATG-9 by genetically interacting with clathrin-associated adaptor complexes such as AP-1, AP-2, and AP180. Our findings are consistent with studies in nonneuronal cells, which demonstrated that the AP-1 and AP-2 complexes mediate ATG-9 trafficking between the plasma membrane, the *trans*-Golgi network (TGN),

the recycling endosomes, and the growing autophagosomes [50,52,87,88]. It is noteworthy that in both *cla-1* and *Synaptojanin/unc-26* mutants, sorting defects of ATG-9 result in ATG-9-containing vesicles abnormally colocalizing onto subsynaptic foci. Clusters of ATG-9-containing vesicular structures have also been observed in vertebrate cells defective for autophagy [89], suggesting that clustering of ATG-9 vesicles, such as those seen for *cla-1* mutants, might be a hallmark of defective autophagy. Consistent with this interpretation, we observe that mutations in early autophagy protein EPG-9 also result in abnormal accumulation of ATG-9 in synaptic foci, which were enhanced by *cla-1(L)* mutants, further underscoring the relationship between ATG-9 sorting at the synapse and autophagy.

Sorting of ATG-9 at synapses is genetically separable from the sorting of synaptic vesicle proteins. Similar to synaptic vesicles, ATG-9 vesicles are transported to synapses via the canonical synaptic vesicle kinesin UNC-104/KIF-1A and undergo exo-endocytosis in an UNC-13/Munc13- and UNC-26/Synaptojanin-dependent manner. However, we hypothesize that ATG-9-containing vesicles probably represent a distinct subpopulation of vesicles at the synapse, for 3 reasons: First, only a small amount of ATG-9 was found in the synaptic vesicle fractions via mass spectrometry [90–92], suggesting that ATG-9 does not localize to all vesicles at synapses. Second, our cell biological studies reveal that while ATG-9 localizes to presynaptic sites, it only partially colocalizes with synaptic vesicle proteins [24] (Fig 2A–2D), consistent with the existence of distinct vesicle pools. Third, in both fibroblasts and nerve terminals, vertebrate ATG9A does not coassemble into synaptophysin-positive vesicle condensates, consistent with it undergoing differential sorting relative to synaptic vesicle proteins [93]. While further biochemical studies on the composition of ATG-9-containing vesicles are required to better understand the relationship between ATG-9 vesicles and synaptic vesicles in neurons, our cell biological studies suggest that they belong to distinct subpopulation of vesicles at the synapse. Moreover, while we do not fully understand the biochemical interactions of CLA-1L resulting in ATG-9 sorting, our in vivo genetic studies suggest a model of how the synaptic machinery could cooperate with the autophagy pathway in regulating local synaptic autophagy. We favor a model in which AP-1 adaptor complex and the F-BAR protein syndapin I (SDPN-1) mediate trafficking of ATG-9 to a transient sorting station from which AP2-AP180 complexes facilitate clathrin-mediated ATG-9 vesicle budding (Fig 10).

Protein turnover (e.g., via autophagy) is correlated with activity states of synapses and is necessary for optimal synaptic function [94,95]. It is interesting that in *cla-1(L)* mutants, only activity-induced autophagy, but not baseline autophagy, is affected in the AIY interneurons. We speculate that other molecules play a redundant role in supporting baseline autophagy. Proteins like CLA-1L could be necessary to boost synaptic autophagy for the degradation of damaged synaptic components under high activity states and via sorting of ATG-9. We hypothesize that the sorting could potentiate ATG-9 role as a lipid scramblase in the nucleation of the growth of the autophagosome isolation membrane [96–99], perhaps by the activation of the scramblase activity via exo-endocytosis, and the subsequent transport of lipids necessary for autophagosome biogenesis [100,101]. In summary, we propose a model whereby active zone proteins, like CLA-1L, which bridge the exocytic active zone with the endocytic periactive zone, could regulate ATG-9 sorting to modulate this activity-dependent presynaptic autophagy.

## Materials and methods

### *C. elegans* strains and genetics

*C. elegans* Bristol strain worms were raised on NGM plates at 20˚C using OP50 *Escherichia coli* as a food source [102]. Larva 4 (L4) stage hermaphrodites were examined. For a full list of strains used in the study, please see S2 Table.

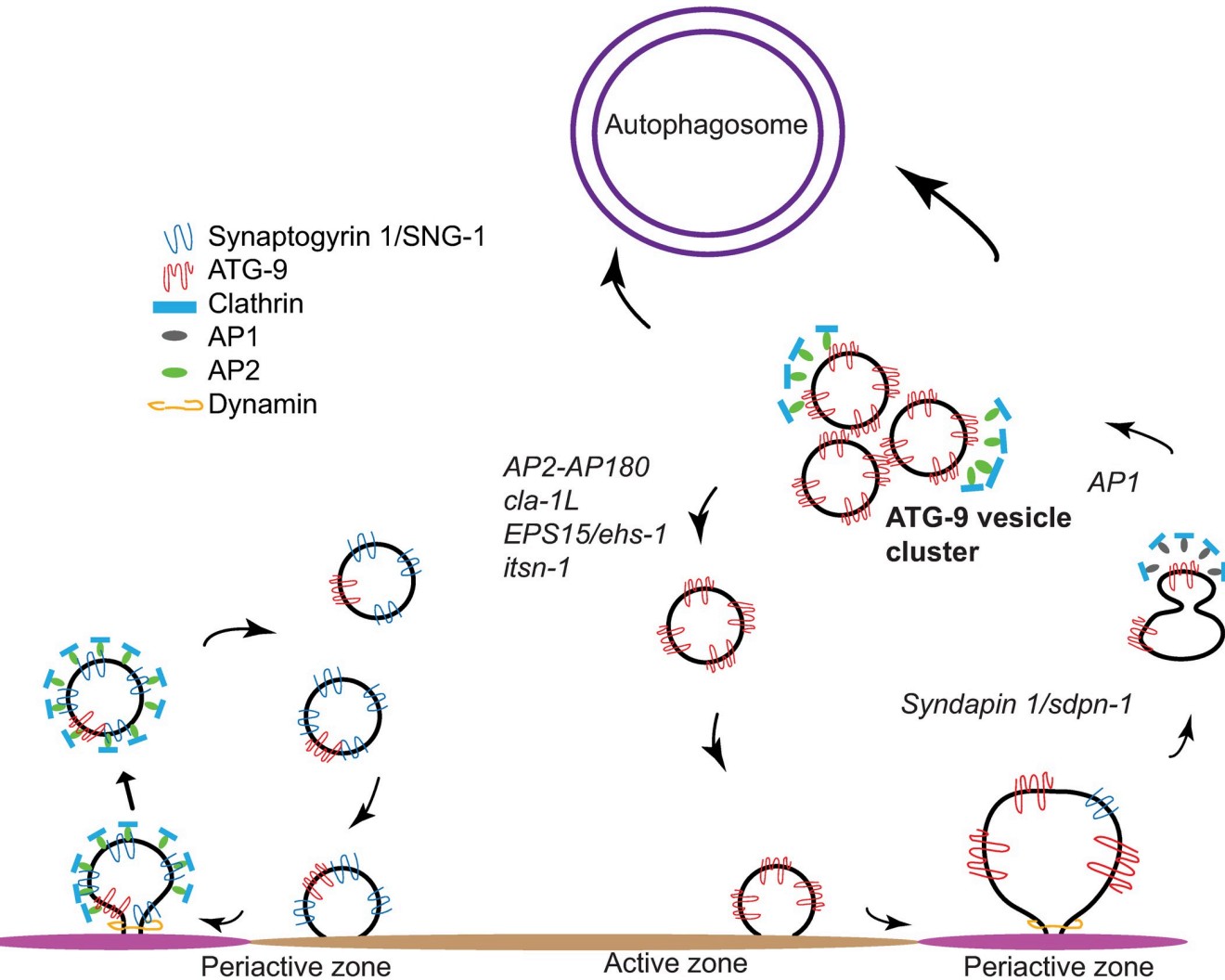

**Fig 10. Cartoon diagram representing the genetic relationships between ATG-9 trafficking, the synaptic vesicle cycle, and synaptic autophagy.** At Zone 2 of AIY, both synaptic vesicles and ATG-9 vesicles undergo exo-endocytosis [24]. Our data are consistent with ATG-9 undergoing distinct sorting pathways and displaying distinct phenotypes than those seen for synaptic vesicle proteins. We propose a model whereby ATG-9 is sorted by the adaptor complex AP1 to intracellular endocytic intermediates (symbolized here by "ATG-9 vesicle cluster"). CLA-1L, together with presynaptic endocytic proteins that reside in the periactive zone, such as EHS-1 and ITSN-1, as well as the adaptor complexes such as AP-2 and AP180, are necessary for sorting of ATG-9 from endocytic intermediates. Mutations in the active zone gene *cla-1L* result in abnormal accumulation of ATG-9 into endocytic intermediates and defects in activity-dependent autophagosome formation.

## Molecular biology and transgenic lines

Expression clones were made in the pSM vector [103]. The transgenic strains (0.5 to 50 ng/μl) were generated using standard injection techniques and coinjected with markers Punc122:: GFP (15 to 30 ng/μl), Punc122::dsRed (15 to 30 ng/μl), or Podr-1::rfp (15 to 30 ng/μl). UNC-101, mouse AP1 mu1, and mouse AP2 mu isoform1 were PCR amplified from *C. elegans* and mouse cDNA (PolyATtract mRNA Isolation Systems, Promega and ProtoScript First Strand cDNA Synthesis Kit, NEB). The plasmid sequences with annotations are included in S1–S5 Plasmids files and can be viewed via ApE (https://jorgensen.biology.utah.edu/wayned/ape/), a free, multiplatform application for visualizing, designing, and presenting biologically relevant DNA sequences.

## Forward genetic screen, SNP mapping, and whole-genome sequencing (WGS)

*Cla-1(ola285)* was isolated from a visual forward genetic screen designed to identify mutants with abnormal localization of ATG-9::GFP in the interneuron AIY. *OlaIs34* animals, expressing ATG-9::GFP and mCherry::RAB-3 in AIY interneurons, were mutagenized with ethyl methanesulfonate (EMS) as described previously [102], and their F2 progenies were visually screened under A Leica DM 5000 B compound microscope with an oil objective of HCX PL APO 63×/1.40–0.60 for abnormal ATG-9::GFP localization. SNP mapping [32] was used to map the lesion corresponding to the *ola285* allele to a 6.6- to 11.3-Mbp region on chromosome IV. WGS was performed at the Yale Center for Genome Analysis (YCGA) and analyzed on www.usegalaxy.org using "Cloudmap Unmapped Mutant workflow (w/ subtraction of other strains)" as described [33,34]. The *ola285* allele was sequenced by using Sanger sequencing (Genewiz), and the genetic lesion confirmed as a single T-to-A nucleotide substitution in Exon 15 of *cla-1L* that results in a missense mutation I5753N. Complementation tests were performed by generating *ola285/cla-1(ok560) trans*-heterozygotes. The *ola285* allele failed to complement *cla-1(ok560)*.

## N-terminal endogenous labeling of CLA-1L

To endogenously tag CLA-1L at the N-terminus in *cla-1(ola285)* mutants, a CRISPR protocol [25,104] was used to create *cla-1(ola506[gfp::cla-1L]);cla-1(ola285)*.

## Cell autonomy of CLA-1L

A CRISPR protocol [105,106] was used to create *cla-1 (ola324)*, in which 2 loxP sites were inserted into 2 introns of *cla-1L* [25]. Cell-specific removal of CLA-1L in AIY was achieved with a plasmid driving the expression of Cre cDNA under the AIY-specific *mod-1* promoter fragment [107].

## Cell autonomy and cell-specific rescues

The ATG-9 phenotype in *unc-101(m1);cla-1(ola285)* was suppressed by cell specifically expressing the *C. elegans* cDNA of *unc-101(m1)/AP1μ1*C and the murine cDNA of *AP1μ1*. Cell-specific expression was achieved using AIY-specific ttx-3G promoter fragment [108]. Cell-specific expression of the murine cDNA of *AP2μ* in AIY did not suppress the ATG-9 phenotype in *unc-101(m1);cla-1(ola285)*.

## Confocal imaging

Hermaphrodite animals of larval stage 4 (L4) were anesthetized using 10 mM levamisole (Sigma-Aldrich) or 50 mM muscimol in M9 buffer, mounted on 2% to 5% agar pads and imaged by using a 60× CFI Plan Apochromat VC, NA 1.4, oil objective (Nikon) on an Ultra-View VoX spinning-disc confocal microscope (PerkinElmer). Images were processed with Volocity software. Maximum-intensity projections presented in the figs were generated using Fiji (NIH) for all the confocal images. Quantifications were performed on maximal projections of raw data. Note that the ATG-9 subsynaptic foci in *cla-1(ola285)* were imaged with a different (lower exposure) confocal setting from the wild-type control (higher exposure), to avoid saturating the signal in *cla-1 (ola285)* animals and at the same time to maximize ATG-9 signal (below saturation) in wild-type animals. The Index of ATG-9 punctum at Zone 2 of AIY for each genotype was consistent across different confocal settings (S1E–S1I Fig).

## Electron microscopy

High-pressure freezing, freeze substitution, and sectioning were all performed as previously described in [24,109–111]. AIY Zone 2 was identified based on the anatomical landmarks described in the original *C. elegans* connectome and recent Zone 2 reconstruction [24,31].

For immuno-EM, sections of 50 nm were collected on nickel slot grids covered with Formvar (EMS). Grids were incubated at 20˚C on 50 μl droplets of 0.05 M glycine PBS for 5 min, 1% BSA and 1% CWFS gelatin in PBS for 20 min, anti-GFP rabbit polyclonal (1:20 in 0.3% BSA and 0.3% CWFS gelatin in PBS, ab6556 Abcam) overnight at 4˚C and then 60 min at 20˚C, 6 PBS washes over 30 min, Protein A Gold conjugated to 10 nm gold (1:75 in 0.3% BSA and 0.3% CWFS gelatin in PBS, University Medical Center Utrecht) for 60 min, 6 PBS washes over 30 min, 2% glutaraldehyde in PBS for 5 min, 3 water washes for 10 s. After drying, grids were post-stained in 2% uranyl acetate for 4 min, and lead citrate for 1 min.

Images were acquired on TALOS L120 (Thermo Fisher) equipped with a Ceta 4k × 4k CMOS camera. For serial sections, images were aligned in z using the TrakEM2 plugin in FIJI [112].

## Quantification and statistical analyses

**Quantifications of penetrance and expressivity.** A Leica DM500B compound fluorescent microscope was used to visualize and screen the worms in the indicated genetic backgrounds. Penetrance was scored blindly as either wild type (distributed throughout the Zone 2 synaptic region) or mutant (localized into subsynaptic foci) phenotypes for ATG-9 or SNG-1 at Zone 2 of AIY. Mutant phenotype was defined as one or more subsynaptic foci of ATG-9:: GFP or SNG-1::GFP at Zone 2 of AIY. For each genotype, at least 40 animals were scored. The Y axis of graphs is named as "% of animals with ATG-9 (or SNG-1) puncta at Zone 2 of AIY." Statistics for penetrance quantification was determined using two-tailed Fisher's exact test. Error bars represent 95% of confidence intervals.

In addition, we validated our scoring results by quantifying the expressivity of the phenotypes. Specifically, confocal micrographs of around 15 representative worms for each genotype were acquired using a spinning-disc confocal microscope (PerkinElmer) within the dynamic range of fluorescence (avoiding saturated pixels). Fluorescence values for each AIY Zone 2 were obtained after background subtraction by drawing a freehand line using Fiji along the long axis of Zone 2. The fluorescence peak values and trough values were acquired via the Profile Plot function. Subsynaptic enrichment index was then calculated as $(F_{peak}-F_{trough})/F_{trough}$. The Y axes of the graphs are named as "Index of ATG-9 (or SNG-1) punctum at Zone 2 of AIY." To compare penetrance across different genotypes presented in different graphs of the study, quantifications of wild type from Fig 1M and *cla-1(ola285)* mutant from Fig 4J were included as references in the following figures: Figs 1M, 4J, 5M, 6G, 8P, S1B, S1G, and S4E. To compare expressivity across different genotypes presented in different graphs of the study, quantifications of wild-type and *cla-1(ola285)* mutant from Fig 1L were included as references in the following figures: Figs 1L, 4K, 5N, 6H, 7I, S1L, and S7G (we clearly stated this in the figure legends). Similarly, quantifications of percentage and expressivity for *unc-11(47)* from Fig 5M and 5N were included as references for Figs 5M, 6G, and S4E and Figs 5N, 6H, and 7I respectively.

**Mean intensity of SYD-2 at the presynaptic Zone 2 of AIY neurons.** To measure the level of SYD-2 at presynaptic regions, we obtained the fluorescent value using Fiji as indicated above. All settings for the confocal microscope and camera were kept identical to compare the intensity of SYD-2 between the wild-type and *cla-1(ok560)* mutants. The same ROI was drawn for all micrographs analyzed. The mean fluorescent value of SYD-2 was measured by Fiji.

**Mean intensity of CLA-1L at the nerve ring region.** To measure the level of CLA-1L at the nerve ring region, we drew the same ROI in all micrographs analyzed and the mean fluorescent value of CLA-1L was measured by Fiji. All settings for the confocal microscope and camera were kept identical to compare the intensity of CLA-1L between the wild-type and *cla-1(ola285)* mutants.

**Colocalization analysis and analyses of endogenous CLA-1 localization at the presynaptic regions.** The Coloc2 plugin of Fiji was used to measure the Pearson correlation coefficient for colocalization between ATG-9::GFP and CHC-1::BFP, or ATG-9::GFP and SNG-1::BFP, both in *cla-1(ola285)* mutants. ROI was drawn to include the entire Zone 2 in all Z-stacks. The Coloc2 plugin of Fiji was also used to measure the Pearson correlation coefficient for colocalization between APT-4::mCh and CLA-1::GFP or APT-4::mCh and GFP::CLA-1L.

To be able to rigorously examine the relationship of CLA-1 to subsynaptic domains, we chose a neuronal system in which active zone and periactive zone positions had been well characterized both by EM and light microscopy: the posterior dorsal nerve cord [31,71]. Note that APT-4::mCh is expressed in a subset of neurons in the dorsal nerve cord, driven by the punc-129 promoter, while CLA-1::GFP and GFP::CLA-1 are expressed panneuronally (so green puncta can be present where there are no magenta puncta; Fig 8D–8N and Methods). Therefore, ROI was drawn based on the outline of APT-4::mCh puncta for both CLA-1::GFP and GFP::CLA-1L to compare the colocalization between CLA-1::GFP or GFP::CLA-1L with APT-4::mCh in the neurons that express APT-4::mCh.

**Activity-dependent autophagy.** To measure the synaptic autophagosomes in AIY, *olaIs35* animals expressing eGFP::LGG-1 and cytoplasmic mCherry in AIY interneurons in the wild-type, *cla-1(ola285)*, and *cla-1(ok560)* mutant backgrounds were raised in 20˚C and then shifted to 25˚C for 4 h as described [24] to alter the activity state of the AIY interneuron [80]. LGG-1 puncta in the neurite of AIY was scored under a Leica DM 1048 5000B compound microscope as described [12].

**Quantification of transmission EM and immuno-EM.** Quantifications were performed using the TrakEM2 plugin in FIJI. Plasma membranes, endosomes, and dense projections areas in AIY were manually traced as area lists. Synaptic vesicles, dense core vesicles, and immunogold particles were manually marked as ball objects. The layers were scaled in the z-dimension (40 nm for transmission EM, 50 nm for immuno-EM) before making 3D models. Active zone length was calculated by counting the number of continuous sections, which had a dense projection present in AIY and then multiplying by 40 nm (morphology section thickness). The number of synaptic vesicles or dense core vesicles per cross section was calculated by taking the number of marked synaptic vesicles or dense core vesicles within AIY's cell membrane per z-layer. The endosomes area was calculated by measuring the total marked endosomes area in the sections that possess endosomes. The relationship between the area measurements and the volume is the thickness of the section, which is 40 nm. Because this number is constant for all sections, the volume is directly proportional to all our area measurements. The Analyze Particles tool was used in FIJI to count the immunogold particles, which labelled the ATG-9::GFP. AIY's cell membrane was manually traced and then a threshold set to only show the darkest 3% on the image. The Analyze Particles tool was set to detect particles 40 to 120 $nm^2$ with a circularity of 0.8 to 1.0. The ATG-9 density (immunogold particles per area) was calculated by dividing the number of ATG-9 immunogold particles by the area of AIY neuron in the corresponding sections.

**Statistical analyses.** Statistical analyses were conducted with Prism 7 software and reported in the figure legends. Briefly, two-tailed Fisher's exact test was used to determine statistical significance of categorical data such as the penetrance of ATG-9 phenotype. For continuous data, such as Index of ATG-9 punctum, two-tailed unpaired Student *t* test or ordinary

one-way ANOVA with Tukey's post hoc analysis was used. For LGG-1 scoring, since the number of LGG-1 puncta for each condition did not pass D'Agostino and Pearson normality test, the nonparametric analysis Kruskal–Wallis with Dunn's multiple comparisons test was used. The error bars represent either 95% confidence interval or standard deviation (SD), as indicated in the figure legends, along with the *p*-values.

## Protein sequence alignment of the AP adaptor complex *μ* subunit

The sequence of *C. elegans* unc-101/AP-1μ1 (K11D2.3) was acquired from WormBase (https://wormbase.org). The sequences of murine AP-1μ1 (NP_031482.1) and murine AP-2μ (NP_001289899.1) were acquired from NCBI (https://www.ncbi.nlm.nih.gov/protein/?term=). Sequence alignment of worm UNC-101/AP1μ1, mouse AP1μ1, and mouse AP2μ was generated using Tcoffee in Jalview 2.10.5 [113].

## Supporting information

**S1 Data. Source data for graphs in this paper.**
(XLSX)

**S1 Table. Annotated list of alleles from forward genetic screen.** List of all alleles identified in 3 semiclonal forward genetic screens and categorized by phenotypic class. We focused our study on the *ola285* allele due to its phenotype (affects ATG-9 localization but does not affect synaptic vesicle protein localization in Zone 2 synapses) and the higher penetrance and expressivity of its phenotype (see Fig 1).
(DOCX)

**S2 Table. Key resources table.**
(DOCX)

**S1 Fig. Examination of the active zone protein SYD-2 in *cla-1(L)* mutants and of ATG-9 distribution in *cla-1(wy1048)* null allele.** (**A**) Schematic of *cla-1* gene, with different protein isoforms. The *ok560* allele specifically affects the long protein isoform, while *wy1048* allele affects all CLA-1 protein isoforms. (**B**) Quantification of the percentage of animals displaying ATG-9 subsynaptic foci at AIY Zone 2 in the indicated genotypes. The data used to quantify the percentage of animals displaying ATG-9 subsynaptic foci in wild-type are the same as those in Fig 1M (explained in Methods). Error bars represent 95% confidence interval. ****$p < 0.0001$ by two-tailed Fisher's exact test. The number on the bars indicates the number of animals scored. (**C**) Mean intensity of GFP::CLA-1L (WT) and GFP::CLA-1L (I5753N) in the worm nerve ring. Error bars show standard deviation (SD). ****$p < 0.0001$ by two-tailed unpaired Student *t* test. Each dot in the scatter plot represents a single animal. (**D**) Quantification of the percentage of animals displaying ATG-9 subsynaptic foci at AIY Zone 2 in the indicated genotypes. The data used to quantify the percentage of animals displaying ATG-9 subsynaptic foci in wild-type are the same as those in Fig 1M (explained in Methods). Error bars represent 95% confidence interval. ****$p < 0.0001$ by two-tailed Fisher's exact test. The number on the bar indicates the number of animals scored. (**E-H**) Distribution of ATG-9::GFP at Zone 2 of AIY in wild type (WT) (**E**), WT with lower exposure setting (**F**), *cla-1 (ola285)* (**G**), and *cla-1(ola285)* with lower exposure setting (**H**). Arrows (in **G** and **H**) indicate abnormal ATG-9 foci. Note that for this study, the ATG-9 subsynaptic foci in *cla-1(ola285)* were imaged with a different (lower exposure) confocal setting from the wild-type control (higher exposure), to avoid saturating the signal in *cla-1 (ola285)* animals and at the same time to maximize ATG-9 signal (below saturation) in wild-type animals. (**I**) Quantification of the index of ATG-9 punctum (ΔF/F) at Zone 2 of AIY for indicated conditions. The data used to

quantify the index of ATG-9 punctum ($\Delta$F/F) in wild-type and *cla-1(ola285)* mutants are the same as those in Fig 1L (explained in Methods). Error bars show standard deviation (SD). "NS" (not significant), **$p < 0.01$ by ordinary one-way ANOVA with Tukey's multiple comparisons test. Each dot in the scatter plot represents a single animal. Note that the index was consistent for ATG-9 in wild-type animals with different imaging confocal settings and was smaller than that in *cla-1 (ola285)* animals. Scale bar (in **E** for **E-H**), 1 μm. Data for S1B, S1C, S1D and S1I Fig can be found in S1 Data.
(EPS)

**S2 Fig. ATG-9 colocalizes better with clathrin than with the integral synaptic vesicle membrane protein SNG-1.** (**A**) Quantification of the index of SNG-1::GFP punctum ($\Delta$F/F) at Zone 2 of AIY in wild-type (WT) and *cla-1(ola285)* mutants. Error bars show standard deviation (SD). "NS" (not significant) by two-tailed unpaired Student *t* test. Each dot in the scatter plot represents a single animal. (**B**) Pearson correlation coefficient for colocalization between ATG-9::GFP and SNG-1::BFP, or between ATG-9::GFP and BFP::CHC-1, both in *cla-1 (ola285)* mutants. Error bars show standard deviation (SD). ****$p < 0.0001$ by two-tailed unpaired Student *t* test. Each dot in the scatter plot represents a single animal. Data for S2A and S2B Fig can be found in S1 Data.
(EPS)

**S3 Fig. Dense core vesicles and endosomal structures in the electron micrographs of wild-type and *cla-1(ola285)* mutants did not reveal major differences.** (**A**, **B**) Distribution of SYD-2::GFP at the synaptic Zone 2 and Zone 3 regions of AIY in wild-type (**A**) and *cla-1 (ok560)* (**B**) animals. The dashed boxes highlight the presynaptic Zone 2 of AIY examined in this study. (**C**) Mean intensity of SYD-2 at AIY Zone 2 in wild-type (WT) and *cla-1(ok560)* mutants. Error bars show standard deviation (SD). "NS" (not significant) by two-tailed unpaired Student *t* test. Each dot in the scatter plot represents a single animal. (**D**, **E**) Electron microscopy of the Zone 2 region in wild-type (**D**) and *cla-1(ola285)* mutant animals (**E**). Same as Fig 2I and 2J, without annotations. (**F**) Quantification of dense core vesicles in the AIY neurons (AIYL: AIY on the left side; AIYR: AIY on the right side) of 1 wild-type and 1 *cla-1 (ola285)* mutant. Error bars represent standard deviation (SD). ****$p < 0.0001$ by ordinary one-way ANOVA with Tukey's multiple comparisons test. Each dot in the scatter plot represents a single section. (**G**) Measurement of endosome area in the AIY neurons (AIYL: AIY on the left side; AIYR: AIY on the right side) of 2 wild-type and 1 *cla-1(ola285)* mutant. Error bars represent standard deviation (SD). *$p < 0.05$ by ordinary one-way ANOVA with Tukey's multiple comparisons test. Each dot in the scatter plot represents a single section. (**H**) Total number of endosomes in the AIY neurons of wild-type and *cla-1(ola285)* mutants. Error bars represent standard deviation (SD). "NS" (not significant) by two-tailed unpaired Student *t* test. *n* = 4 for wild-type and *n* = 2 for *cla-1(ola285)* mutants. Scale bar (in **A** for **A** and **B**), 5 μm; (in **D** for **D** and **E**), 500 nm. Data for S3C, S3F, S3G and S3H Fig can be found in S1 Data.
(EPS)

**S4 Fig. The AP-1 and AP-2 adaptor complexes mediate presynaptic trafficking of ATG-9.** (**A-D**) Distribution of ATG-9::GFP at Zone 2 of AIY in wild-type (WT) (**A**), *unc-101(m1);cla-1(ola285)* mutants with mouse AP2μ cDNA cell specifically expressed in AIY (**B**), *dpy-23 (e840)/AP2μ* (**C**), and *unc-101(m1);dpy-23(e840)* (**D**) mutant animals. ATG-9 subsynaptic foci are indicated by the arrow (in **C**). (**E**) Quantification of the percentage of animals displaying ATG-9 subsynaptic foci at AIY Zone 2 in the indicated genotypes. The data used to quantify the percentage of animals displaying ATG-9 subsynaptic foci in wild-type and *cla-1(ola285)* mutants are the same as those in Fig 4J; the data used in *unc-11(e47)* are the same as those in

Fig 5M (explained in Methods). Error bars represent 95% confidence interval. "NS" (not significant), $^*p < 0.05$, $^{**}p < 0.01$, $^{****}p < 0.0001$ by two-tailed Fisher's exact test. The number on the bars indicates the number of animals scored. Black asterisks indicate comparison between each group with the wild-type control. Blue asterisks indicate comparison between two specific groups (highlighted with brackets). (**F**) Sequence alignment of *C. elegans* UNC-101/AP1μ1, mouse AP1μ1, and mouse AP2μ, generated using Tcoffee in Jalview (Waterhouse and colleagues [113]). Although both AP-1 μ1 and AP-2μ share similarity, *C. elegans* UNC-101/AP1μ1 is more similar to the mouse AP1μ1 (Query Cover: 100%; Percentage Identity: 74%) than the mouse AP2μ (Query Cover: 98%; Percentage Identity: 41%). Scale bar (in **A** for **A**-**D**), 1 μm. Data for S4E Fig can be found in S1 Data.
(EPS)

**S5 Fig. Cartoon diagrams of the genetic interactions, and model, in this study.** (**A**) Mutants for CLA-1L, AP-2, and AP180 adaptor complexes display similar ATG-9 phenotypes at synapses and are necessary for resolving ATG-9-containing foci (clathrin-rich endocytic intermediates). Note that the pink-filled symbols represent the abnormal ATG-9 foci in mutants for CLA-1L, AP-2, and AP180 adaptor complexes. (**B**) Cartoon diagram of the genetic interactions, and model displaying the genetic interactions of CLA-1L with SDPN-1/syndapin 1 and the AP-1 adaptor complex, which suppress the ATG-9 phenotypes observed for *cla-1(ola285)*, *AP-2*, or *AP180* mutants, consistent with roles for SDPN-1/syndapin 1 and the AP-1 upstream of the formation of the abnormal ATG-9 foci (similar to what was observed for exocytosis mutants in Fig 3, which also suppressed ATG-9 phenotypes in *cla-1(ola285)* mutants).
(EPS)

**S6 Fig. Schematics of the strategies for endogenously tagging CLA-1 at C-terminus or N-terminus via CRISPR.** (**A**) Schematics of the strategy for endogenously tagging CLA-1 at C-terminus via CRISPR. FLP-on-GFP (let-858 3′ UTR flanked by FRT sites followed by GFP) was inserted at the common C-terminus of CLA-1 (shared by CLA-1L, CLA-1M, and CLA-1S protein isoforms; see Xuan and colleagues [25]). FLPase driven by the Prab-3 promoter is expressed panneuronally to induce expression of CLA-1::GFP in an endogenous manner (see Fig 8). (**B**) Schematics of endogenously tagging CLA-1L at N-terminus via CRISPR (Xuan and colleagues [25]). GFP was inserted at the unique N-terminus of CLA-1L (see Fig 8).
(EPS)

**S7 Fig. CLA-1L genetically interacts with the endocytic proteins at the periactive zone to regulate ATG-9 trafficking.** (**A**-**F**) Distribution of ATG-9::GFP at Zone 2 of AIY in wild-type (WT) (**A**), *cla-1(ola285)* (**B**), *ehs-1(ok146)* (**C**), *ehs-1(ok146);cla-1(ola285)* (**D**), *itsn-1(ok268)* (**E**), and *itsn-1(ok268);cla-1(ola285)* (**F**) mutant animals. (**G**) Quantification of the index of ATG-9 punctum (ΔF/F) at Zone 2 of AIY in the indicated genotypes. The data used to quantify the index of ATG-9 punctum (ΔF/F) in wild-type and *cla-1(ola285)* mutants are the same as those in Fig 1L (explained in Methods). Error bars show standard deviation (SD). "NS" (not significant), $^*p < 0.05$ by ordinary one-way ANOVA with Tukey's multiple comparisons test. Each dot in the scatter plot represents a single animal. Scale bar (in **A** for **A**-**F**), 1 μm. Data for S7G Fig can be found in S1 Data.
(EPS)

**S8 Fig. Disrupted ATG-9 trafficking in *cla-1(ok560)* mutants is associated with a deficit in activity-induced autophagosome formation.** (**A**) Quantification of the average number of LGG-1 puncta in the AIY neurites at 20˚C and at 25˚C for 4 h in wild-type (WT) and *cla-1 (ok560)* mutants. Error bars represent 95% confidence interval. "NS" (not significant), $^{***}p < 0.001$ by Kruskal–Wallis test with Dunn's multiple comparisons test. The number on

the bars indicates the number of animals scored. Data for S8A Fig can be found in S1 Data.
(EPS)

**S1 Plasmids. Please use ApE to open (https://jorgensen.biology.utah.edu/wayned/ape/).**
(STR)

**S2 Plasmids. Please use ApE to open (https://jorgensen.biology.utah.edu/wayned/ape/).**
(STR)

**S3 Plasmids. Please use ApE to open (https://jorgensen.biology.utah.edu/wayned/ape/).**
(STR)

**S4 Plasmids. Please use ApE to open (https://jorgensen.biology.utah.edu/wayned/ape/).**
(STR)

**S5 Plasmids. Please use ApE to open (https://jorgensen.biology.utah.edu/wayned/ape/).**
(STR)

## Acknowledgments

We thank Jihong Bai (Basic Sciences Division, Fred Hutch) and Kang Shen (Department of
Biology, Stanford University) for providing strains and constructs. We thank Lin Shao
(Department of Neuroscience, Yale University) for assistance with image quantification and
statistics. We thank Josh Hawk for providing mouse cDNA. We thank Center for Cellular and
Molecular Imaging, Electron Microscopy Facility, the Neuroscience EM core at Yale Medical
School for assistance with the work presented here, and Morven Graham, Leslie Gunther-
Cummins, Yumei Wu, Irina Kolotuev, Leslie Gunther-Cummins, David Hall, Maike Kittel-
mann, and Szi-chieh Yu for advice on immunoelectron microscopy experiments. We thank
current and past members of the Colón-Ramos lab for help, advice, and insightful comments
on the project. We also thank Andrea Stavoe, Ian Gonzalez, Mia Dawn, Peri Kurshan, Janet
Richmond, and Pietro De Camilli for assistance and comments on the project. We thank the
Caenorhabditis Genetics Center (funded by NIH Office of Research Infrastructure Programs
P40 OD010440) for *C. elegans* strains.

## Author Contributions

**Conceptualization:** Zhao Xuan, Sisi Yang, Daniel A. Colón-Ramos.

**Data curation:** Zhao Xuan, Benjamin Clark, Daniel A. Colón-Ramos.

**Formal analysis:** Zhao Xuan, Benjamin Clark.

**Funding acquisition:** Daniel A. Colón-Ramos.

**Investigation:** Zhao Xuan, Sisi Yang, Daniel A. Colón-Ramos.

**Methodology:** Zhao Xuan, Sisi Yang, Benjamin Clark, Laura Manning.

**Supervision:** Zhao Xuan, Daniel A. Colón-Ramos.

**Validation:** Zhao Xuan, Sisi Yang.

**Visualization:** Zhao Xuan, Sisi Yang, Benjamin Clark, Sarah E. Hill.

**Writing – original draft:** Zhao Xuan, Daniel A. Colón-Ramos.

**Writing – review & editing:** Zhao Xuan, Sisi Yang, Benjamin Clark, Sarah E. Hill, Daniel A.
    Colón-Ramos.

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
