## [Editor Report · Decision Letter 0]

15 Nov 2022

Hi Daniel,

As I mentioned in my email from the other day, we need you to do one more step to complete your submission for your manuscript entitled "The active zone protein Clarinet regulates synaptic sorting of ATG-9 and presynaptic autophagy" by providing the metadata that is required for full assessment. 

To this end, please login to Editorial Manager where you will find the paper in the 'Submissions Needing Revisions' folder on your homepage. Please click 'Revise Submission' from the Action Links and complete all additional questions in the submission questionnaire. To provide the metadata for your submission, please Login to Editorial Manager (https://www.editorialmanager.com/pbiology) within two working days, i.e. by Nov 17 2022 11:59PM.

Once your full submission is complete, your paper will undergo a series of checks. After your manuscript has passed the checks we'll discuss the study and the portable-peer review comments with an Academic Editor. 

Kind regards,

Kris

Kris Dickson, Ph.D., (she/her)

Neurosciences Senior Editor/Section Manager

PLOS Biology

kdickson@plos.org

---

## [Decision Letter · Decision Letter 1]

9 Jan 2023

Dear Dr Colon-Ramos,

Thank you for your patience while your manuscript "The active zone protein Clarinet regulates synaptic sorting of ATG-9 and presynaptic autophagy" was peer-reviewed at PLOS Biology. It has now been evaluated by the PLOS Biology editors, an Academic Editor with relevant expertise, and by several independent reviewers - all of whom were specifically asked to evaluate the work in light of the prior comments you'd received at Dev Cell and your point-by-point response to that prior feedback. (Note - we have 4 reviewers on this as, in our attempts to find a 3rd reviewer, we invited a few people at once and two agreed at the same time.)

Based on the reviewers' overall positive responses, we are likely to accept this manuscript for publication, provided you satisfactorily address the points raised by the reviewers with some rewriting and reorganization (including, amongst other changes, moving Supplemental Figure S1D into main Fig 1). Please also make sure to address the data and other policy-related requests listed below my signature. These must be fully addressed before we can proceed further.

***Please also provide a blurb which, if the paper is accepted, will be included in our weekly and monthly Electronic Table of Contents (eTOCs), sent out to readers of PLOS Biology. This blurb may also be used to promote your article on social media. The blurb should be about 30-40 words long and is subject to editorial changes. It should, without exaggeration, entice people to read your manuscript, should not be redundant with the title and should not contain acronyms or abbreviations. For examples, view our author guidelines: https://journals.plos.org/plosbiology/s/revising-your-manuscript#loc-blurb

We expect to receive your revised manuscript within two weeks. 

*Published Peer Review History*

*Press*

Sincerely,

Kris

Kris Dickson, Ph.D., (she/her)

Neurosciences Senior Editor/Section Manager,

kdickson@plos.org,

PLOS Biology

DATA POLICY:

Note that we do not require all raw data. Rather, we ask that all individual quantitative observations that underlie the data summarized in the figures and results of your paper be made available. While we've noted you've listed that all data is “fully available within the manuscript and supporting documents, we did not see any summary data files provided as yet.

Please provide summary data files in one of the following formats:

2) Deposition in a publicly available repository. Please also provide the accession code or a reviewer link so that we may view your data before publication. Deposition in a public repository should also include a description field when uploading your files using the following format verbatim: S1 Data, S2 Data, etc. Multiple panels of a single or even several figures can be included as multiple sheets in one excel file that is saved using exactly the following convention: S1_Data.xlsx (using an underscore).

Current figure numbering: Fig1L,M,R; Fig2M,N; Fig3E; Fig4J,K; Fig5M,N; Fig6G,H; Fig7I; Fig8O,P; Fig9D,I

Supplemental current figure numbering: Fig1B,E,H; Fig2A,B; Fig3A,B; Fig4E; Fig6I; Fig7;

Please also include the following:

1) A noted above, Please move Fig1SD into the main body of the paper, adding this panel to Figure 1 as requested by Reviewer 2 and by Reviewer 4 in their follow-up comments as well. Please also discuss these findings in more detail as the reviewers have requested.

2) Please modify your figure legends to include statements indicating where your summary data can be found, saying something like “The summary data for Figure X can be found in the supplemental files as FILE NAME.”

3) Please also ensure your supplemental data files have legends. 

DATA NOT SHOWN?

- Please note that per journal policy, we do not allow the mention of "data not shown", "personal communication", "manuscript in preparation" or other references to data that is not publicly available or contained within this manuscript. Please carefully check your submission for any such occurrences and either remove mention of these data or provide figures presenting the results and the data underlying the figure(s).

Reviewer remarks:

Do you want your identity to be public for this peer review?

Reviewer #1: No

Reviewer #2: No

Reviewer #3: No

Reviewer #4: No

Reviewer #1: The article "The active zone protein Clarinet regulates synaptic sorting of ATG-9 and presynaptic autophagy" by Xuan et al. shows that the active zone protein, Clarinet, regulates the activity-dependent sorting of ATG-9 at presynaptic terminals. This study provides new understanding to the field regarding how autophagosome proteins may be sorted at synapses, and suggests which proteins at the active zone and peri-active zone play key roles in this sorting. The authors addressed the major concerns raised by reviewers in the first version of the manuscript submitted to Dev Cell. I think the paper is suitable for publication in PLoS Biology after addressing the following minor comments:

1. Supplemental figure 1C, D: Note that there was a problem with the images in this figure in one of our pdf viewers, such that only half of the image was visible. We were able to see the image in another viewer, but it may be worth a double check before publication.

2. Figure 2I and 2J: The blue line is too thick and obscures structures near the plasma membrane, such as the active zone. The authors should make the line thinner.

3. Figure 2N and S3, which were added during revisions, are not adequately discussed in the text. 

235 "Quantification of the presence of synaptic vesicles, dense core

236 vesicles and endosomal structures in the electron micrographs of cla-1(ola285)

237 and wild-type animals did not reveal major differences that could account for the

238 observed phenotypes of ATG-9 (Figure 2 I-N; Figure S3A-B).

The graph indicates that there are significant changes in synaptic vesicle and dense core vesicle number in the mutant, and it's unclear why the left and right neurons should have opposite phenotypes in the mutant. Does this discrepancy suggest that this experiment is not adequately powered to make quantitative conclusions? Are the cross-sections of similar area and if they are not, would it be better to normalize to area? The authors should discuss this result rather than glossing it over.

4. I do not agree with the author's interpretation of the immuno-EM data. They wrote:

255 Together, 

256 our findings suggest that the ATG-9 phenotype in cla-1(ola285) results from 

257 differences in the sorting of ATG-9-containing vesicular structures at the synapse. 

I think the immuno-EM data confirms (and does not add much to) what they observe with the light microscopy which is that the ATG-9 positive vesicles are more clustered at presynaptic terminals in the cla-1 mutant compared to controls. I don't agree this data indicates that ATG-9 is sorted differently because they do not detect any morphological alterations that would suggest the nature of the ATG-9-associated compartment has changed in cla-1(ola285). These data only show that the distribution, not the sorting, of the ATG-9 vesicles is altered in cla-1(ola285) compared to controls. I think there are many explanations that could account for changes in distribution such as altered trafficking, mobility, recruitment to condensates etc. Though the experiments they mention later in the paper support the idea that the sorting of ATG-9 is disrupted, the altered distribution and disrupted sorting may be two separate phenomena.

5. Figure 8: I think this experiment really clearly shows that the C-terminus of CLA-1 localizes to the active zone while the N-terminus localizes to the peri-active zone. However, this experiment was not performed in the AIY interneurons. The authors should explain why they performed this single experiment in neurons of the posterior dorsal nerve cord instead of the AIY interneurons. 

Reviewer #2: This paper by Xuan et al follows up a recently published study from the same lab (Yang et al, 2022), which reports the biogenesis of ATG-9 containing vesicles from Golgi via AP-3 and how ATG-9 containing vesicles at AIY neuron synapses undergo activity-dependent exo-endocytosis. Yan et al described a phenotype that in endocytosis mutants such as unc-26/synaptojanin, ATG-9 forms clathrin-rich subsynaptic foci. Here, the authors identified, through genetic screening, a missense mutation in the active zone protein cla-1L that causes ATG-9 synaptic foci phenotype resembling unc-26 mutants. They performed EM reconstruction of AIY synapses and found that overall presynaptic active zone and vesicles in the AIY neurons are largely unaltered in the cla-1L mutants. Immuno-EM of ATG-9::GFP show altered distribution in cla1L mutants. They show CLA-1L localizes to a presynaptic subdomain periactive zone, different from other cla-1 protein isoforms. They carried out extensive genetic interaction studies between cla-1l and AP1, AP2 and other endosomal sorting proteins. These data led them to conclude that ATG-9 sorting in AIY synapse depends synaptic exo-endocytosis, as well as coordinated action of endosomal sorting machinery. Finally, they show mis-sorting of ATG-9 in cla-1L mutants contributed to activity-induced autophagosome formation at synapses. 

The manuscript presents a large body of work. The AIY synapse EM reconstruction is very impressive. The analysis on CLA-1L and ola285 mutation is careful, and informative about synaptic protein subdomains. While the message that CLA-1L mediated sorting of ATG-9 is dependent on synapse exo-endocytosis sounds similar to that in Yang et al, the differential effects of AP1, AP2, EPS15 and others in ATG-9 sorting make important contribution to understand synaptic endosomal network. A missing puzzle is that the entire study did not touch on how exactly CLA-1-mediated ATG-9 sorting interface with unc-26 or endophilins, even though the similarity of cla-1L to unc-26 in ATG-9 localization phenotype was the starting point of the study and emphasized throughout the manuscript. Some data and conclusions may be clarified and strengthened. 

Specific points: 

Line 198-199, regarding the analysis on cla-1(wy1048): please clarify if the phenotype is only referring to ATG-9 synaptic foci.

Line 272-273 conclusion: 'These results demonstrate that the ATG-9 phenotype in cla-1(ola285) mutants results from defects in ATG-9 sorting upon ATG-9 exo-endocytosis.' It seems that this is based on double mutant analysis using synaptic vesicle exocytosis mutants. How did author draw the link to ATG-9 exo-endocytosis? 

Line 297-298 and Figure 5M-N: the graphs need cla-1(ola285) for static comparison to support the enhanced expressivity of ATG-9 sorting defects. 

Figure S1D: the imaging seems to show GFP::CLA-1L(I5753N) protein. if so, please correct the panel label. This data would be better to be shown in the Figure 1 . The effect of the I5753N mutation on CLA-1L protein is striking. The large size of CLA-1L protein makes biochemical studies difficult. Does protein modeling help shed some insights? 

Pg 11-12, Figure 5E-H and Figure S2B: how does unc-26/synaptojanin and endophilin affect the co-segregation of ATG-9 and CHC-1 in cla-1(ola285)? Similarly, please show data or comment on if AP3 has any role in cla-1L mediated ATG-9 sorting at AIY synapse. 

Figure 6G-H: does sdpn-1; unc-11 show any difference from sdpn-1 alone? does mutation in sdpn-1 suppress ATG-9 phenotypes in unc-11 and cla-1 to the same degree? 

Figure 7I: does unc-101 suppresses cla-1 and unc-11 to same degree? 

Figure 8: the data nicely show N- and C-terminal tagged CLA-1 protein subsynaptic difference in the nerve cord. Since ATG-9 foci phenotype seems to be specific to AIY synapse zone 2, it would be appropriate to know how CLA-1 is localized in AIY, or revise the text to make it clear which synapses are and what the relevance is to ATG-9 sorting

Minor points: 

Line 95: 'sort of', of is extra word?

Line 158: consider to revise the sentence 'The phenotypes for synaptic vesicle proteins and ATG-9 differ in ola285 mutants' as it implies ola285 also affects synaptic vesicle proteins. 

Line 598, a typo: the cla-1L mutation is stated as nonsense mutation. 

Figure 1M: it is unclear why there is no statistic analysis between cla-1(ola285)/cla-1(ok560) with respect to each homozygous mutation. 

Consider to re-organize two supplemental figures to make contents cohesive. 

Figure S1F-H is unrelated to A-E. 

Figure S6 C-D is unrelated to A-B

Figure S5: the cartoon diagrams are nice, but it is unclear what 'OR' is intended for. 

line 1280: 'insets it' means 'insets at'

Reviewer #3: In this manuscript, Xuan et al performed an unbiased genetic screen (by mutagenesis) in C. elegans to identify proteins involved in the autophagosome cycle, and specifically of vesicles incorporating ATG-9. Their main finding is that the active zone protein Clarinet is specifically involved in the recycling of ATG-9 containing vesicles, but does not appear to participate in the recycling of synaptic vesicles. They use mutants of proteins involved in sequential steps of vesicle sorting to illustrate the dependence of cycling and sorting of ATG-9 on clarinet. Specifically, they show that disrupting bulk endocytosis antagonizes the clarinet-dependent deficit in ATG-9 localization, while disruption of sorting of endocytosis intermediates enhances it. Likewise, inhibiting exocytosis also antagonizes the clarinet dependent effect on ATG-9 cycling. Also, ATG-9 colocalizes with clathrin when sorting of endocytosis products is deficient. The authors fuse GFP to both ends of clarinet to show that it extends from the active zone to the endocytosis-active peri-active zone. In this context, they illustrate a cooperative relationship between clarinet and the EPS15/Intersectin1 endocytic scaffolding that couples between active zone exocytosis and periactive zone endocytosis. Finally, they show that the mutation in clarinet reduces activity-dependent autophagosome formation, based on imaging of LGG-1.

This manuscript is a revised version, after it was referred to PLOS biology from Dev Cell. The authors addressed much of the concerns raised by the previous reviewers. Furthermore, they clearly differentiated between the novel findings of this study (the involvement of clarinet in the sorting of ATG-9 containing vesicles) and the findings of a previous study already published by the same group in Neuron concerning the involvement of ATG-9 in SV autophagy.

I don't have major concerns about the manuscript, and I find that it contributes significantly to the advancement of our knowledge on this important topic. Furthermore, the manuscript is written clearly, and the logic behind the experiments is explained amply. 

I do have a few questions for the authors:

1. The authors indicate that clarinet is involved in activity-dependent autophagy and sorting of ATG-9, but that baseline autophagy is not affected. On this basis, wouldn't they expect that the ATG-9 phenotype in clarinet mutants should resolve over extended periods of time? I am puzzled by the fact that the authors observe steady-state ATG-9 puncta, but they indicate that only activity-dependent autophagy is affected by the clarinet mutation. In this context, is there a difference between the number of LGG1 puncta at 20C between WT and clarinet mutants? The authors provided a statistical analysis only for the effect of the shift in temperature but not the effect of the ola285 mutation itself (as far as I could tell).

2. In this respect, do the authors have any EM data relating directly to the autophagosomes themselves? I.e., in their diagrams the authors allude to recycling endosomes or other intermediates and to autophagosomes that should include ATG-9, but do not visualize those directly, at least in the EM data. According to the diagrams (Figure 10 and supplementary figure 5), it could be expected to observe such ATG-9 labeled intermediates, especially in clarinet mutants.

3. The authors claim that clarinet does not affect synaptic vesicle cycling, based on their examination of Rab3 distributions in WT and ola285 mutants. Do they have functional readouts to substantiate this very important claim (besides immunofluorescence analyses)?

4. I could not understand why the authors observed a difference between ola285-AIYL and ola285-AIYR in figure 2. Is the mutation targeted to one but not the other?

Reviewer #4: The manuscript by Xuan et al. examines the novel role of active zone scaffold protein Clarinet/CLA-1 in mediating the sorting of autophagic lipid scramblase protein ATG-9 at presynapses. Using unbiased forward genetic screens of presynaptic ATG-9 localization in the AIY neuron of C. elegans, the authors identify the long isoform of CLA-1 as a regulator of ATG-9 presynaptic distribution/sorting. Through a series of genetic experiments combined with confocal and electron microscopy, the authors suggest a model in which ATG-9 is endocytosed and sorted into endocytic intermediates via proteins including Syndapin-1 and the AP-1 complex, while CLA-1L and the AP-2 and AP180 complexes sort ATG-9 out of these endocytic intermediates and into exocytosis-ready vesicles. Additionally, the authors show that activity-induced autophagosome formation at presynapses in AIY are impaired in CLA-1L mutants. 

While not providing a complete mechanism, this manuscript does offer strong evidence for a role for CLA-1L as a mediator of ATG-9 sorting at synapses, and requires only minor clarifications/revisions:

1. Confocal imaging parameters: The authors explain in the methods section that ATG-9::GFP was imaged using different exposure settings between WT and cla-1(ola285) animals. While the decision to use these different parameters is explained in the author response to a previous reviewer, all validation should be clearly communicated in a supplemental figure that includes representative images of both genotypes at both exposures, and with comparisons of the index of ATG-9 puncta between all four conditions. Providing this information will help more convincingly make the argument that the relationship between the indices of ATG-9 distribution remains consistent, even when imaging at different exposures between genotypes.

2. In Figure 2N, each point is reported to represent a single section. Further information about the number of neurons (n) represented in each condition should be reported. Additionally, the significant difference between AIY-right in cla-1(ola285) mutants and the other genotypes is reported but not commented on in the text. Is this finding relevant to the fluorescence images presented, which are not labeled as AIY-left or -right in any figures? 

3. In the quantification of Supplemental Figure 3, the average endosome area is displayed for each cross section, but further details about the total number and volume of endosomes present in each genotype would be helpful to understand whether there is any effect on the endosomal pathway. Indeed, in the model provided in Figure 10, the intracellular endocytic intermediates are displayed in two forms (endosomes or vesicles) with "or" written between them. This makes the model more difficult to follow, and the information gleaned from the multiple EM reconstructions in this work should offer some insight into the morphology of these intermediates. 

4. In Figure 8, it is confusing and unclear why the overall colocalization between CLA-1 and APT-4 is so low, and only being a C. elegans researcher and digging deep into the strain information was it possible to understand that APT-4 is only being expressed in a subset of neurons in the dorsal nerve cord (under the Punc-129 promoter), while CLA-1::GFP-FLP-on is expressed pan-neuronally (through Prab-3 > Flpase) and GFP::CLA-1 is constitutively expressed pan-neuronally. A better experimental design would be to drive APT-4 from a pan-neuronal promoter when comparing with GFP::CLA-1L strains, in order to compare colocalization of CLA-1L with APT-4 in all neurons along the dorsal nerve cord. Additionally, the C-terminally tagged CLA-1::GFP-FLP-on could be expressed in the same subset of neurons as APT-4 by expressing flippase driven by Punc-129. At the minimum, further clarification of the fact that APT-4 is expressed in only a subset of neurons in the dorsal nerve cord, and what this means for the colocalization analysis, should be provided in the text. However, the localization of CLA-L is a central point in the paper, and the proposed experiments are quite simple, and would more convincingly demonstrate the point being made, especially to a non-worm reader. 

5. At first glance in Figure 9A-C, the inset boxes appear to be additional regions within the image, rather than magnifications of the boxed regions below. Perhaps boxing the magnified insets in a solid line would be clearer.

---

## [Editor Report · Decision Letter 2]

8 Feb 2023

Dear Dr Colon-Ramos,

Thank you for the submission of your revised Research Article "The active zone protein Clarinet regulates synaptic sorting of ATG-9 and presynaptic autophagy" for publication in PLOS Biology. On behalf of my colleagues and the Academic Editor, Cagla Eroglu, I am pleased to say that we can in principle accept your manuscript for publication, provided you address any remaining formatting and reporting issues. These will be detailed in an email you should receive within 2-3 business days from our colleagues in the journal operations team; no action is required from you until then. Please note that we will not be able to formally accept your manuscript and schedule it for publication until you have completed any requested changes.

PRESS

Sincerely, 

Kris

Kris Dickson, Ph.D., (she/her)

Neurosciences Senior Editor/Section Manager

PLOS Biology

kdickson@plos.org